# Identification of an analgesic lipopeptide produced by the probiotic *Escherichia coli* strain Nissle 1917

Teresa Pérez-Berezo[1], Julien Pujo[1], Patricia Martin[1,2], Pauline Le Faouder[3], Jean-Marie Galano[4], Alexandre Guy[4], Claude Knauf[1], Jean Claude Tabet[5], Sophie Tronnet[1], Frederick Barreau[1], Maud Heuillet[6], Gilles Dietrich[1], Justine Bertrand-Michel[3], Thierry Durand[4], Eric Oswald[1,2] & Nicolas Cenac[1]

Administration of the probiotic *Escherichia coli* strain Nissle 1917 (EcN) decreases visceral pain associated with irritable bowel syndrome. Mutation of *clbA*, a gene involved in the biosynthesis of secondary metabolites, including colibactin, was previously shown to abrogate EcN probiotic activity. Here, we show that EcN, but not an isogenic *clbA* mutant, produces an analgesic lipopeptide. We characterize lipoamino acids and lipopeptides produced by EcN but not by the mutant by online liquid chromatography mass spectrometry. One of these lipopeptides, C12AsnGABA*OH*, is able to cross the epithelial barrier and to inhibit calcium flux induced by nociceptor activation in sensory neurons via the GABA$_B$ receptor. C12AsnGABA*OH* inhibits visceral hypersensitivity induced by nociceptor activation in mice. Thus, EcN produces a visceral analgesic, which could be the basis for the development of new visceral pain therapies.

[1] IRSD, Université de Toulouse, INSERM, INRA, INP-ENVT, Université de Toulouse 3 Paul Sabatier, 31024 Toulouse, France. [2] CHU Toulouse, Hôpital Purpan, Service de bactériologie-hygiène, 31024 Toulouse, France. [3] MetaToulLipidomics Facility, INSERM UMR1048, 31432 Toulouse, France. [4] Institut des Biomolécules Max Mousseron IBMM, UMR 5247 CNRS, Université de Montpellier-ENSCM, 34093 Montpellier, France. [5] Sorbonne Université, UPMC Univ Paris 06, CNRS, Institut Parisien de Chimie Moléculaire (IPCM), 75005 Paris, France. [6] LISBP, Université de Toulouse, CNRS, INRA, INSA, 31077 Toulouse, France. Teresa Pérez-Berezo, Julien Pujo and Patricia Martin contributed equally to this work. Eric Oswald and Nicolas Cenac jointly supervised this work. Correspondence and requests for materials should be addressed to N.C. (email: nicolas.cenac@inserm.fr)

rritable bowel syndrome (IBS) is a functional gastrointestinal disorder characterized by recurrent episodes of abdominal pain/discomfort and bowel habit changes (e.g., constipation, diarrhea)[1]. With a global prevalence of ~11%[1], IBS constitutes one of the most common conditions leading to gastroenterological referral, and results in a considerable disease burden. While the pathophysiology of IBS is not fully understood, visceral hypersensitivity (VH; enhanced sensitivity of the intestinal wall to local stimuli) has been proposed as a key mechanism underlying abdominal pain, one of the most debilitating and most troublesome symptoms of this disorder[2–4]. Current treatments for IBS are mainly symptoms orientated; however, the overall efficacy is low and there are no drugs specifically approved for abdominal pain[5]. Thus, selective pharmacological tools targeting VH may be considered a suitable therapeutic approach for visceral pain treatment and development of novel IBS therapies.

Data from clinical research suggest that certain probiotic bacterial strains have the potential to modulate abdominal pain in IBS[6–9]. Nonetheless, these data differ considerably among studies due to the probiotic bacterial strains used for the treatment and the heterogeneity of IBS groups included. Moreover, the mechanisms of action responsible for the claimed therapeutic effects differ from one strain to another.

*Escherichia coli* Nissle 1917 (EcN) is the active component of Mutaflor® (Ardeypharm GmbH, Herdecke, Germany), a probiotic drug licensed in several countries for the treatment of multiple intestinal disorders[10]. Clinical trials have shown EcN to be effective for the treatment of abdominal pain in IBS patients[11,12], but very little is known about the specific mechanisms through which EcN exerts the ascribed analgesic effects. EcN is known to harbor a genomic island, named *pks*, which carries a cluster of genes that enables the synthesis of hybrid peptide polyketides and especially a genotoxin called colibactin (Fig. 1)[13]. Colibactin is a structurally uncharacterized polyketide (PK)—non-ribosomal peptide (NRP) that is thought to arise from a prodrug called precolibactin, which has also not been fully structurally elucidated[14,15]. This toxin is produced by a complex biosynthetic machinery involving the sequential action of proteins ClbA to ClbS[16]. The core machinery consists of three PK synthases (PKS), three NRP synthetases (NRPS), and two hybrids PKS-NRPS[16]. The machinery also employs additional maturation proteins, smaller enzymes with modules of PKS enzyme, an efflux pump, a resistance protein, and a putative regulatory protein. ClbA a phosphopantetheinyl transferase (PPTase) essential for the activation of the NRPS and PKS enzymes[16]. ClbA is mandatory for the biosynthesis of colibactin[13], but is also involved in the biosynthesis of other bioactive metabolites such as the siderophores enterobactin, salmochelin, and yersiniabactin[17].

Following activation by ClbA, the initiating NRPS ClbN uses Asn as a substrate to generate N-myristoyl-D-Asn (Fig. 1). The NRPS-PKS assembly line continues the synthesis of precolibactin compound(s) using malonyl-coA and different amino acids[18–20] as substrates. The precolibactin is then cleaved by peptidase ClbP to liberate colibactin and N-myristoyl-D-Asn (C14Asn*OH*; Fig. 1)[18,21,22].

Surprisingly, although colibactin was shown to be a bona fide virulence factor and a putative carcinogenic agent[13], this genotoxin is also produced by EcN. The probiotic activity of EcN can apparently not be dissociated from its genotoxic activity, since inactivation of *clbA* required for the activation of the NRPS and PKS enzymes leading to colibactin production also attenuates the probiotic activity of EcN in experimental colitis[23]. A possible explanation for the dual role of colibactin in EcN may be that the *pks* island codes for additional bioactive compounds distinct from colibactin and involved in the probiotic activity[16,24]. This hypothesis has been recently reinforced by the structural characterization of several colibactin pathway-dependent small molecules[14,15,22]. Hence, the identification and functional characterization of new molecules derived from the colibactin encoding hybrid PKS-NRPS biosynthetic gene clusters may help to decipher some of the mechanisms, supporting the capacity of EcN to modulate abdominal pain, and thereby allowing the design of novel analgesic agents devoid of genotoxic properties.

Here, we use liquid chromatography coupled to electrospray source (ESI) with tandem high-resolution mass spectrometry (LC/HRMS and LC-HRMS/MS) to identify a metabolite encoded by the *pks* island that shows anti-nociceptive properties in vitro and in vivo. While an increased number of small molecules derived from the colibactin encoding hybrid PKS-NRPS biosynthetic gene clusters have been described[18,19,22], this study characterizes a non-genotoxic bioactive metabolite. This visceral analgesic produced by probiotic bacteria may represent a promising therapeutic agent in visceral pain.

## Results

**Identification of lipoamino acid and lipopeptides produced by EcN.** In order to characterize the lipids potentially implicated in probiotic properties of EcN, we performed a comparative lipidomic analysis by LC-HRMS of lipids extracted from the wild-type probiotic strain (EcNwt) and an isogenic mutant for *clbA* that has lost its probiotic activity (EcNΔ*clbA*) in a model of colitis[23]. The total ion chromatograms (TIC) obtained from the ESI were compared to characterize compounds with a relative higher concentration in EcNwt compared to EcNΔ*clbA*. As the relative intensity of the peak eluted at 15.33 min constitutes more

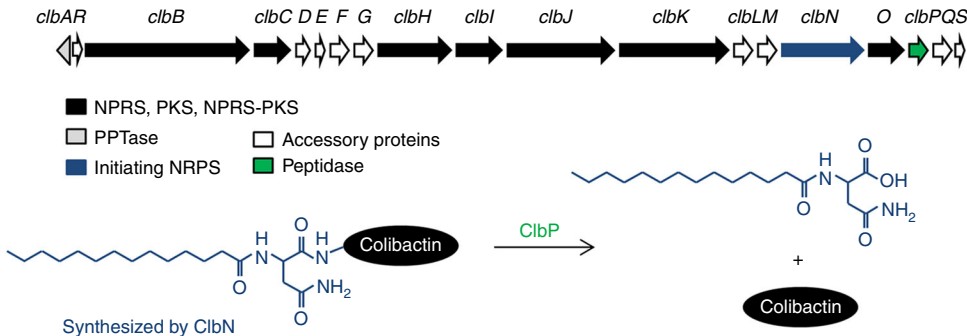

**Fig. 1** Schematic representation of the *pks* genomic island. This cluster of genes encodes the enzymes and accessory proteins (arrows) necessary for synthesis of precolibactin in *E. coli*. Precolibactin is cleaved by a peptidase encoded by *clbP* (green arrow) into active colibactin and a cleavage product, mainly C14Asn*OH*, synthesized by the initiating NPRS encoded by *clbN* (blue arrow)

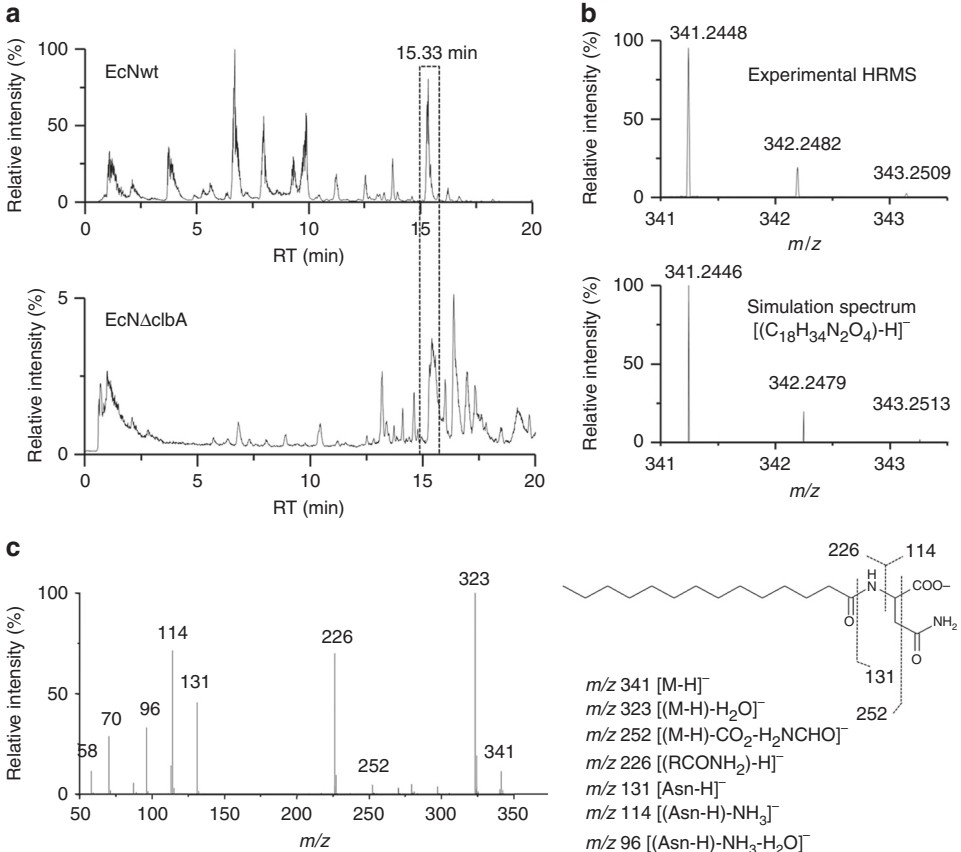

**Fig. 2** Characterization of C14-Asparagine (C14Asn*OH*) by LC-HRMS. **a** TIC of a lipidic extract of EcNwt pellet (up) and EcN∆clbA pellet (down). **b** Natural isotopic distribution of the deprotonated molecule displayed by the high-resolution mass spectrum zoom obtained for the peak eluted at 15.33 min in both TIC (top) and natural isotopic pattern calculated with the formula $[(C_{18}H_{34}N_2O_4)-H]^-$. Analogous natural isotopic patterns and similar m/z ratios measured and simulated for the monoisotopic $[(^{12}C_{18}{}^1H_{34}{}^{14}N_2{}^{16}O_4)-H]^-$ ion and for the $[(^{13}C_nC_{18-n}{}^1H_{34}{}^{14}N_2{}^{16}O_4)-H]^-$ (with n = 1 and 2) ions (within an accuracy of 0.6 ppm). **c** Product ion spectrum acquired via HCD (NCE = 35%) of the carboxylate anion [M–H]⁻ (m/z 341) generated in electrospray from the LC peak eluted at 15.33 min

than 80% of the relative intensity in EcNwt, and only 2.5% in EcN∆clbA, we focused our interest on this peak (Fig. 2a). With the accurate m/z 341.2448 ratio obtained we determined an elemental composition of the deprotonated molecule like $[(C_{18}H_{34}N_2O_4)-H]^-$. Then, the natural isotopic profile of the molecular species in the experimental mass spectrum obtained at 15.33 min was compared to its simulation generated from the formula (calculated: m/z 341.2446). These two isotopic profiles were similar and the accuracy of the m/z measurements of monoisotopic peak was 0.6 ppm (Fig. 2b), as well as the m/z measurements of its first major natural isotopic $[(^{13}CC_{17}H_{34}N_2O_4)-H]^-$ peak supporting the proposed elemental composition. Finally, MS/MS analyses were performed under HR measurement conditions by using collisional activation (Fig. 2c). The product ion spectrum (under normalized collision energy (NCE) = 35% conditions) showed ions: (i) at m/z 131.0455, m/z 114.0189, and m/z 96.0078 that are specific product ions[25] from dissociation of the deprotonated asparagine[26] and (ii) anion at m/z 226.2176 corresponding to the deprotonated myristoyl amide side chain[27].

Interpretation of the main product ions (see Supplementary Table 1) of the molecular species at m/z 341.2448 is described in Supplementary Note 1 and Supplementary Fig. 2, which is consistent with the C14-Asparagine lipoamino acid structure. Indeed, the $H_2O$ and $CO_2$ losses take place frequently from amino acids (i.e., corresponding to m/z 323.2335 and m/z 297.2532, respectively). Interestingly, formation of the m/z 252.2334 ion

(Supplementary Fig. 2b) reflects the consecutive $CO_2$ and $HCONH_2$ losses, the free functional groups of the C14-Asparagine lipoamino acid. Furthermore, the m/z 226.2176 fatty amide ion is generated by release of the fumaric acid amide (Supplementary Fig. 2c). On the other hand, the m/z 131.0455 ion is characteristic of the deprotonated asparagine resulting of the C14 alkyl ketene loss (Supplementary Fig. 2d). Thus, with the fragmentation pattern we identified as expected the C14-Asparagine lipoamino acid, which is the cleavage product of precolibactin[18,21,22].

Consequently, the extracted ion chromatogram (EIC) of the diagnostic product ion at m/z 131.0455 obtained from dissociations of the various deprotonated molecular species of the mixture allowed us to identify the CnAsn*OH* molecules with Cn ≠ C14. Based on the mass resolution of the mass spectrometer allowing the accurate m/z measurements and the analysis of product ion spectra, we identified several lipopeptides with different hydrocarbon chain lengths as already published[21,22]. On the other hand, the characterization of other amino acids linked to C-terminus of the C12Asn*OH* and C14Asn*OH* chains through peptidic bond was then performed by the use of detection of the m/z 295.2021 and m/z 323.2335 product ions. They correspond to the formal $[(CnAsnOH-H)-H_2O]^-$ fragment ions (n = 12 and 14), respectively. By this mean, the EICs of these product ions—as reported in Supplementary Figs. 1a, 3a, respectively—allow to determine each m/z values of the parent ions of m/z 295.2021 and m/z 323.2335 corresponding to m/z 426.2980 and m/z 454.3294, respectively. Thus, a common

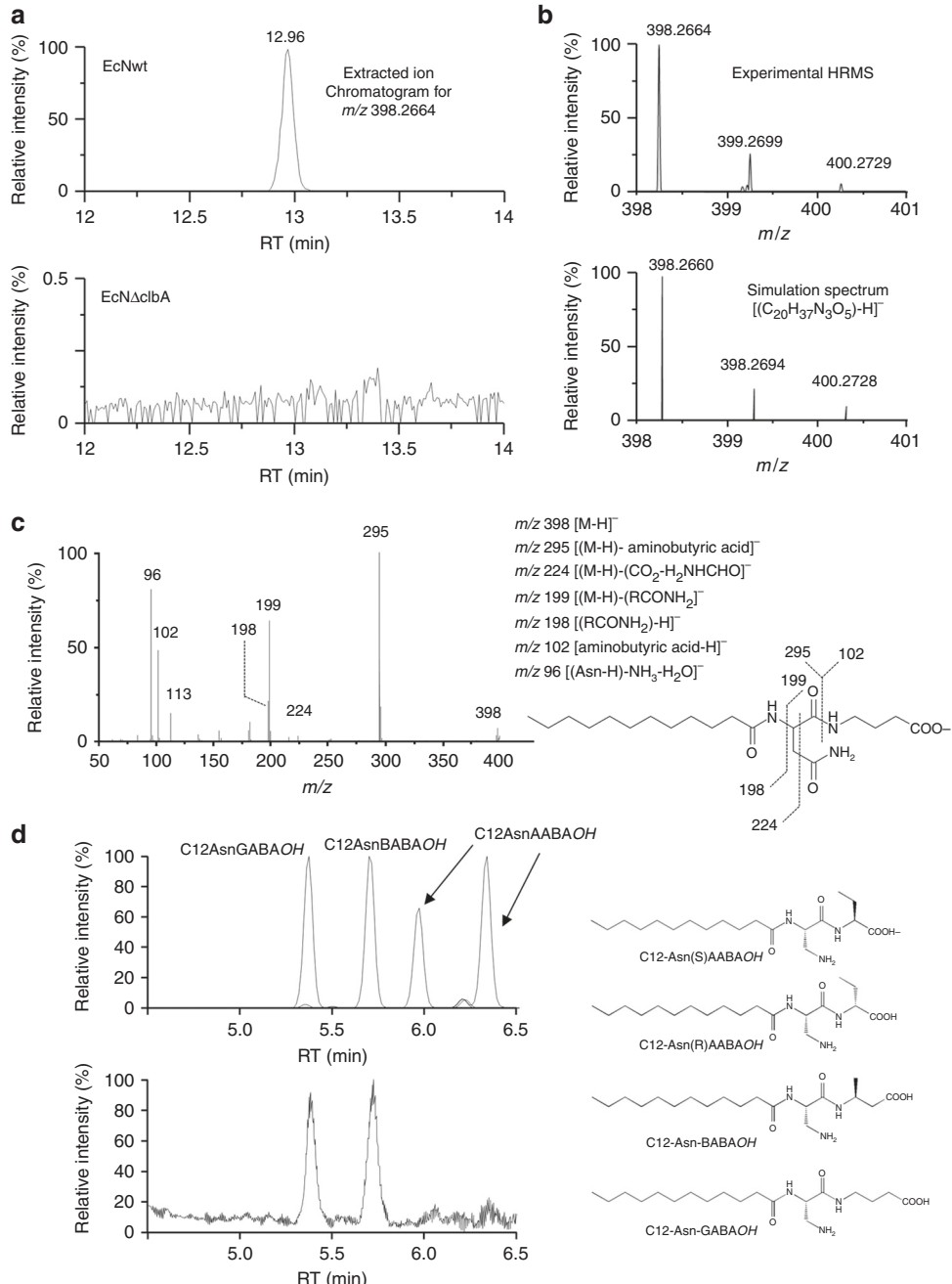

**Fig. 3** Characterization of C12-Asparagine-aminobutyric acid by LC-HRMS. **a** EIC of a lipidic extract of EcNwt (up) and EcNΔclbA pellet (down) for *m/z* 398.2664. No signal was detected in the mutated strain. **b** Natural isotopic distribution of the deprotonated molecule displayed by the high-resolution mass spectrum zoom obtained for the peak eluted 15.33 min in both TIC (top) at 12.96 min in the probiotic strain EIC (top) and natural isotopic pattern calculated with the formula [(C_{20}H_{37}N_3O_5)-H]^-. Analogous natural isotopic patterns and similar *m/z* ratios measured and simulated for the monoisotopic [(^{12}C_{20}^1H_{37}^{14}N_3^{16}O_5)-H]^- ion and for the [(^{13}C_nC_{20-n}^1H_{37}^{14}N_3^{16}O_5)-H]^-]^- (with *n* = 1 and 2) ions (within an accuracy of 1.8 ppm). Same isotopic profile and a mass accuracy of 1.8 ppm were obtained. **c** Product ion spectrum acquired via HCD (NCE = 35%) of the carboxylate anion [M-H]^- (*m/z* 398) generated in electrospray from the LC peak eluted at 12.96 min. **d** Upper panel: chromatogram obtained for the four synthesized standards: C12-Asn-γ-aminobutyric acid (C12Asn-GABA*OH*), C12Asn-(*S*)AABA*OH* and C12Asn-(*R*)AABA*OH*, C12-Asn-α-aminobutyric acid (C12Asn-BABA*OH*) and C12-Asn-α-aminobutyric acid (C12Asn-AABA*OH*) (two diastereoisomers are detected which present similar HCD spectrum with NCE = 35%); lower panel: Chromatogram obtained for the lipid extract of EcNwt pellet

neutral loss (i.e., 131.0959u) occurs from dissociation of these precursor ions to yield the formal [(CnAsn*OH*-H)-H_2O]^- fragment ions, with *n* = 12 or 14, at *m/z* 295.2021 and *m/z* 323.2335. This common neutral release could correspond to a leucine or an isoleucine (Supplementary Figs. 1, 3). The proposed interpretation of formation of these product ions is reported in Supplementary Note 1 and Supplementary Fig. 4a.

We also focused our interest on the ion at *m/z* 398.2663 eluted at 12.96 min (Fig. 3a). The EICs showed that this molecule is present in EcNwt but not in the mutant EcNΔ*clbA* (Fig. 3a). *m/z* ratio accuracy of the monoisotopic peak of [C12AsnLeu*OH*-H]^- at *m/z* 398.2664 displayed by the ESI mass spectrum, under LC-HRMS analysis conditions, allowed to identify its elemental composition, which was confirmed by the simulated natural

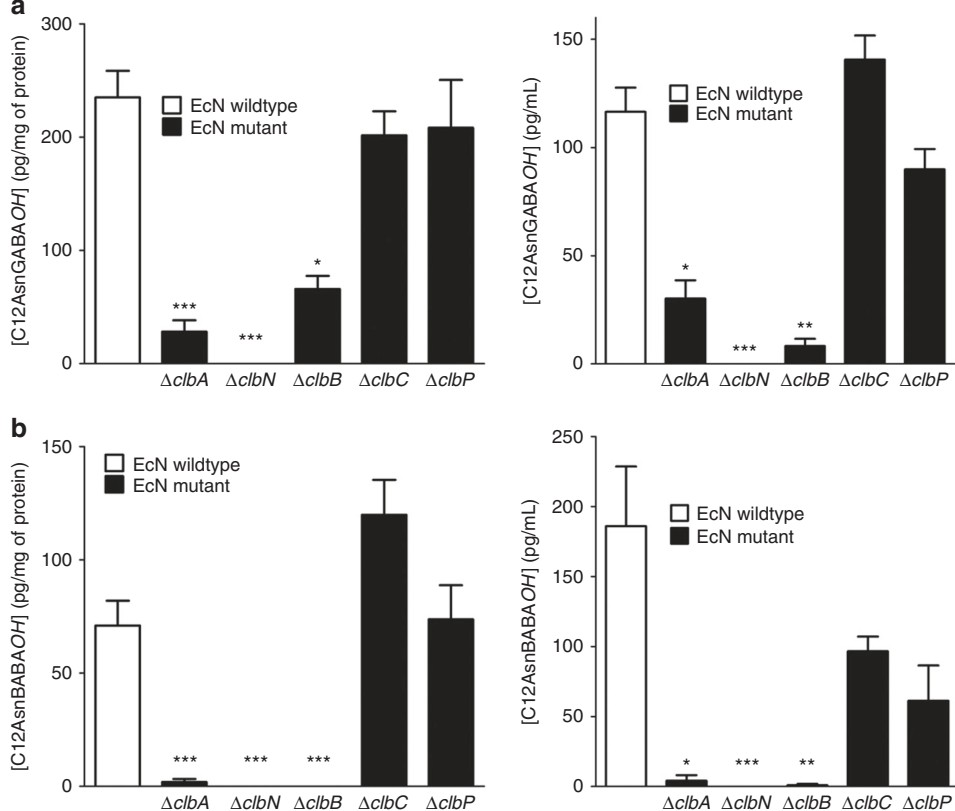

**Fig. 4** Quantification of C12-Asparagine-aminobutyric acid by LC-QQQ. **a** Quantification of C12AsnGABA*OH* in pellets (left panel) and supernatants (right panel) of wild-type and mutated EcN. **b** Quantification of C12AsnBABA*OH* in pellets (left panel) and supernatants (right panel) of wild-type and mutated bacteria. Data are represented as mean ± SEM of two experiments of six independent bacterial cultures per group. Statistical analysis was performed using Kruskal–Wallis analysis of variance and subsequent Dunn's post hoc test. *$p < 0.05$, **$p < 0.01$, ***$p < 0.001$, significantly different from EcNwt

isotopic pattern corresponding to the anion formula $[(C_{20}H_{37}N_3O_5)-H]^-$ (Fig. 3b). Fragmentations displayed by the product ion spectrum enabled us to determine that a butyric acid was linked to a (C12Asn*OH*) lipoamino acid (Fig. 3c). Nevertheless, this characterization did not permit to determine the isomer of butyric acid (branched or not) linked to the Asn moiety. In order to discriminate aminobutyric acid isomers linked to C12Asn*OH*, we synthesized all possible branched ones as L-stereoisomers; C12-Asn-α-aminobutyric (as C12Asn-(S) AABA*OH* and C12Asn-(R)AABA*OH* as a mixture of two isomers due to epimerization of the AABA during the synthesis), C12-Asn-β-aminobutyric (as C12AsnBABA*OH*), and C12-Asn-γ-aminobutyric (as C12AsnGABA*OH*; Fig. 3d, Supplementary Fig. 5, Methods) which were unambiguously characterized by NMR investigations (Supplementary Figs. 6–8). Interestingly, the higher-energy collision dissociation (HCD) spectra recording in high resolution displayed diagnostic differences, which allowed to distinguish these three isomers when the NCE was equal or beyond 35%. Indeed, it appears (Supplementary Table 1) that:

(i) The $m/z$ 312.2276 product ion (Supplementary Figs. 4c and 9c, Supplementary Note 1) is detected only from the BABA structure;

(ii) The series of $m/z$ 155.0822, $m/z$ 138.0557, and $m/z$ 137.0716 product ions characterizes the presence of GABA (Fig. 3c) and AABA (Supplementary Fig. 9f); and

(iii) The product ions at $m/z$ 113.0352 (Supplementary Fig. 9d), $m/z$ 102.0553 (Supplementary Figs. 4a, 9f and Supplementary Note 1), $m/z$ 96.0090 (Fig. 3c, Supplementary Note 1 and Supplementary Fig. 10) are the largest peak at the low

$m/z$ ratio range (<$m/z$ 200) for the BABA, AABA, and GABA substitutions, respectively.

The four synthesized isomers were also analyzed on low-resolution triple quadrupole mass spectrometer coupled online to a liquid chromatography. A 15 min separation method was developed for the separation of each isomer (Fig. 3d). The analysis of bacterial pellets showed the presence of only C12AsnGABA*OH* and C12AsnBABA*OH* (Fig. 3d).

**C12AsnGABA*OH* and C12AsnBABA*OH* synthesis depends on the *pks* island.** *E. coli* K-12 MG1655 was transformed with the bacterial artificial chromosome (BAC) harboring the entire *pks* island (BAC *pks*⁺)[13]. Unlike the MG1655wt, the strain MG1655 +BAC *pks*⁺ was shown to produce C12AsnBABA*OH* and C12AsnGABA*OH* demonstrating the role of *the pks* island in the production of these two compounds (Supplementary Fig. 11). To further elucidate the role of the *pks* island in the synthesis of C12AsnGABA*OH* and C12AsnBABA*OH*, EcNΔ*clbA* and additional isogenic mutants of the EcN strain were generated. These mutants (Δ*clbA*, Δ*clbN*, Δ*clbB*, Δ*clbC*, and Δ*clbP*) were not capable of producing the genotoxin colibactin inducing double-strand breaks in eukaryotic cells, contrary to the parental EcN strain. The lipid metabolite profiles were then characterized in EcN and the mutants, as well as in their culture supernatants. The inactivation of *clbA*, coding for the PPTase, induced a drastic decrease of both C12AsnGABA*OH* and C12AsnBABA*OH* in bacteria and culture supernatants (Fig. 4a, b), confirming the relevance of this gene for the synthesis of these molecules. The *clbN*, *clbB*, *clbC*, and *clbP* genes code for enzymes involved in the first steps and last steps of the biosynthesis of colibactin, namely,

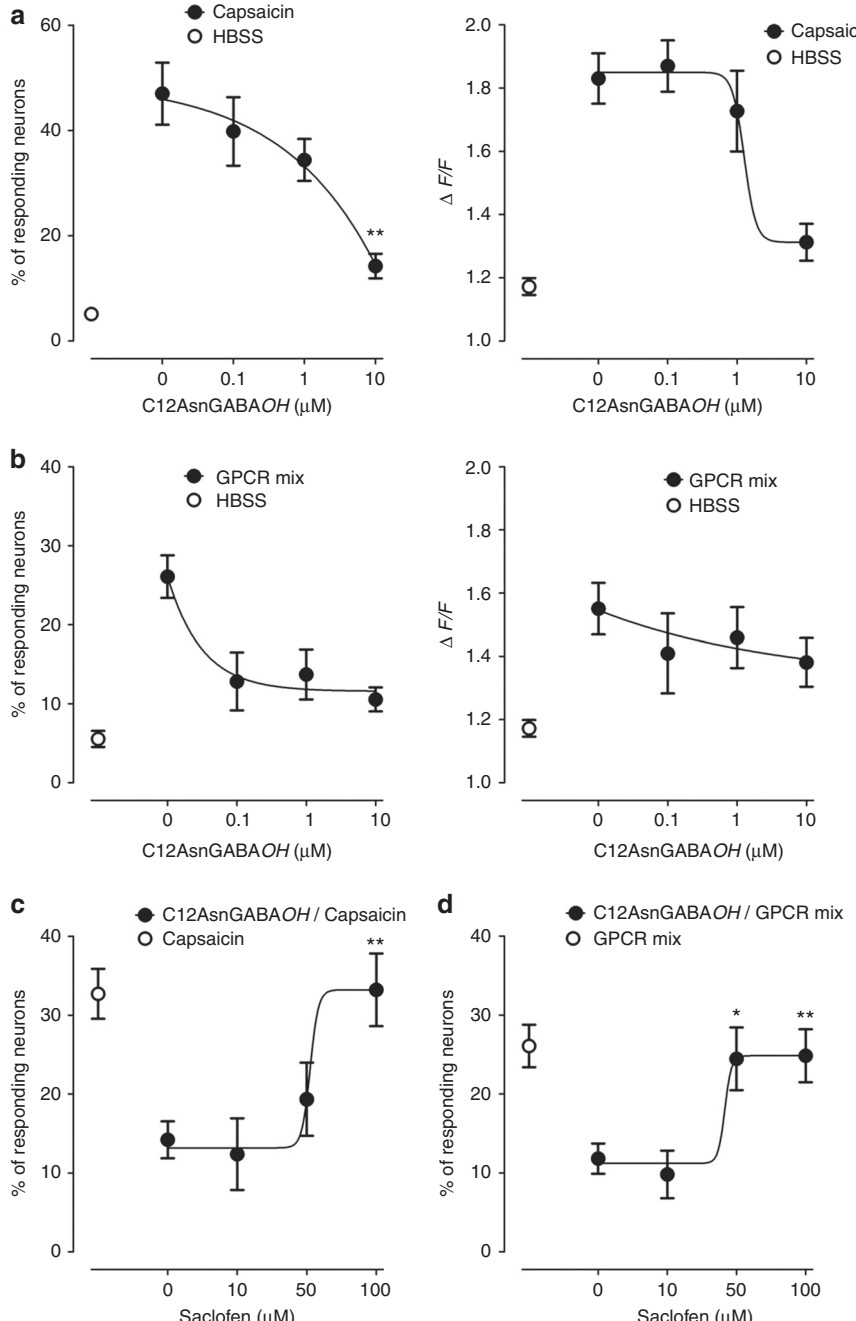

**Fig. 5** C12AsnGABA*OH* inhibits neuronal activation via the GABA_B receptor. Amplitude of intracellular calcium mobilization ($\Delta F/F$; right panel) in mouse sensory neurons and percentage of responding neurons (left panel) pretreated with increasing amounts of C12AsnGABA*OH* (black circle) or vehicle (HBSS; white circle) and treated with capsaicin (125 nM; **a**) or a mix of GPCR agonists (histamine, serotonin, and bradykinin 10 μM each). **b** Data are represented as mean ± SEM; $n = 4$ independent experiments of three wells per condition and 30–80 neurons per well. Statistical analysis was performed using Kruskal–Wallis analysis of variance and subsequent Dunn's post hoc test. *$p < 0.05$, **$p < 0.01$ significantly different from capsaicin or GPCR mix. Percentage of responding neurons pretreated with increasing amounts of saclofen (black circle) or vehicle (HBSS; white circle) and treated with C12AsnGABA*OH* (10 μM) and capsaicin (125 nM; **c**) or a mix of GPCR agonist (histamine, serotonin, and bradykinin 10 μM each). **d** Data are represented as mean ± SEM; $n = 4$ independent experiments of three wells per condition and 30–80 neurons per well. Statistical analysis was performed using Kruskal–Wallis analysis of variance and subsequent Dunn's post hoc test. *$p < 0.05$, **$p < 0.01$ significantly different from C12AsnGABA*OH*/Capsaicin or C12AsnGABA*OH*/GPCR mix

the elongation and the cleavage of the colibactin pro-drug scaffold. Deletion of *clbN* completely abrogated the production and secretion of C12AsnGABA*OH* and C12AsnBABA*OH* in bacteria and their culture supernatants, evidencing the essential role of this gene in the production of these molecules. In a similar manner, *clbB* inactivation abolished the synthesis and secretion of

C12AsnBABA*OH*, and reduced significantly the synthesis and secretion of C12AsnGABA*OH*. On the contrary, mutation of *clbC* (a gene coding for a trans-acyl-transferase PKS that catalyzes an additional round of PK extension, after ClbN and ClbB) did not induce significant changes in the concentration of any of the two molecules in both bacteria and supernatants. Likewise, the *clbP*

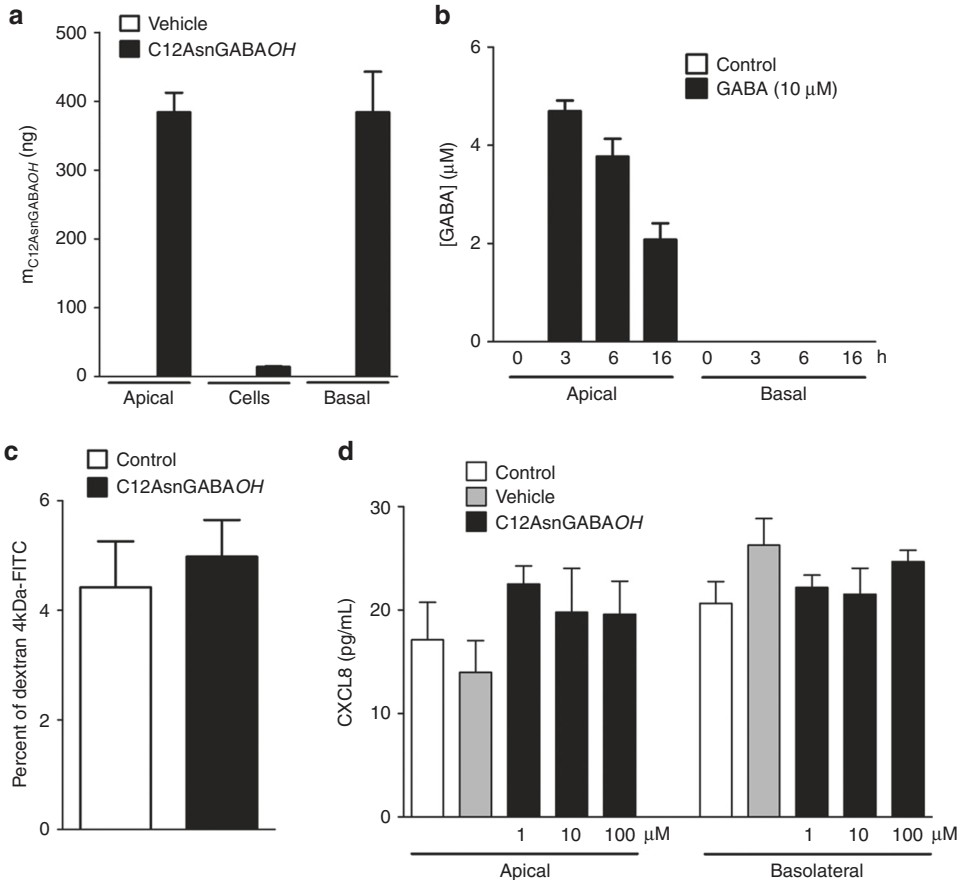

**Fig. 6** C12AsnGABA*OH* crosses the epithelial barrier without modifying epithelial cell physiology. Caco-2 cells were cultivated into transwell chambers. After 24-h treatment with C12AsnGABA*OH* (800 ng) at the apical side, C12AsnGABA*OH* was quantified inside the cells and both the apical and basolateral compartments by LC-MS/MS (**a**); data are represented as mean ± SEM, n = 3 experiments of three wells per condition. Quantification of GABA concentration at the apical and basolateral compartments 3, 6, or 16 h after loading (black bars) or not (white bars) Caco-2 cells with 800 ng of GABA at the apical site (**b**); n = 3 experiments of three wells per condition. Paracellular permeability assessed by the passage of FITC-dextran loaded at the apical side (**c**) and CXCL8 secretion assessed by ELISA (**d**) were determined after 24-h treatment of Caco-2 cells at the apical side with C12AsnGABA*OH* (10 μM); data are represented as mean ± SEM, n = 3 experiments of three wells per condition. Statistical analysis was performed using Kruskal–Wallis analysis of variance and subsequent Dunn's post hoc test

deletion did not modify the synthesis and secretion pattern of any of these molecules, demonstrating that C12AsnGABA*OH* and C12AsnBABA*OH* are not cleavage products. In contrast, the concentration of the cleavage product (C14Asn*OH*) was drastically decreased by *clbP* deletion (Supplementary Fig. 12), as expected. Thus, C12AsnGABA*OH* and C12AsnBABA*OH* are two new molecules dependent on at least three genes of the *clb* biosynthetic genes cluster, confirming the hypothesis that the *pks* island could mediate the formation of compounds with potential probiotic activity[23], in addition to molecules inducing DNA damage[13,28]. As among the aminobutyric acid isomers, γ-aminobutyric (GABA) acid is the primary inhibitory neurotransmitter in the mammalian brain, we hypothesized that C12AsnGABA*OH* was responsible for the anti-nociceptive properties EcN.

**C12AsnGABA*OH* inhibits neuronal activation**. To determine whether C12AsnGABA*OH* or C12AsnBABA*OH* are capable of signaling to sensory nerves, calcium mobilization studies were performed on primary cultures of mouse dorsal root ganglia (DRG) neurons. None of the isomers (10 μM) induced calcium mobilization under basal (unstimulated) conditions (Supplementary Fig. 13). The same experiments were thereafter performed in neurons activated by either an agonist of the receptor

calcium channel TRPV1 (capsaicin) or by a mix of agonists (histamine, serotonin, and bradykinin) for G-protein-coupled receptors (GPCR) implicated in VH. Exposure of neurons to either capsaicin (125 nM) or the mix of GPCR agonists (histamine, bradykinin, serotonin, 10 μM each) induced an increase in calcium flux as shown by the higher % of responding neurons and amplitude of the response (ΔF/F) compared to the vehicle (Fig. 5a, b). The calcium flux increase induced by both nociceptive stimuli was prevented by C12AsnGABA*OH* pretreatment in a dose-dependent manner (Fig. 5a, b), whereas C12AsnBABA*OH* had no effect (Supplementary Fig. 14). Thus, C12AsnGABA*OH* does not induce calcium mobilization in sensory neurons but inhibits neuronal activation induced by pro-nociceptive stimuli in a concentration range similar to GABA alone (Supplementary Fig. 15). To investigate whether the inhibitory effect of C12AsnGABA*OH* was associated to the GABA residue, neurons were treated with saclofen (10, 50, and 100 μM), a competitive antagonist of the GABA_B receptor. Treatment with saclofen abolished the inhibitory effect of C12AsnGABA*OH* against capsaicin and the mix of GPCR agonists in a dose-dependent manner (Fig. 5c, d). Taken together, these results demonstrate that C12AsnGABA*OH* is capable of inhibiting calcium signaling in primary afferents via the GABA_B receptor.

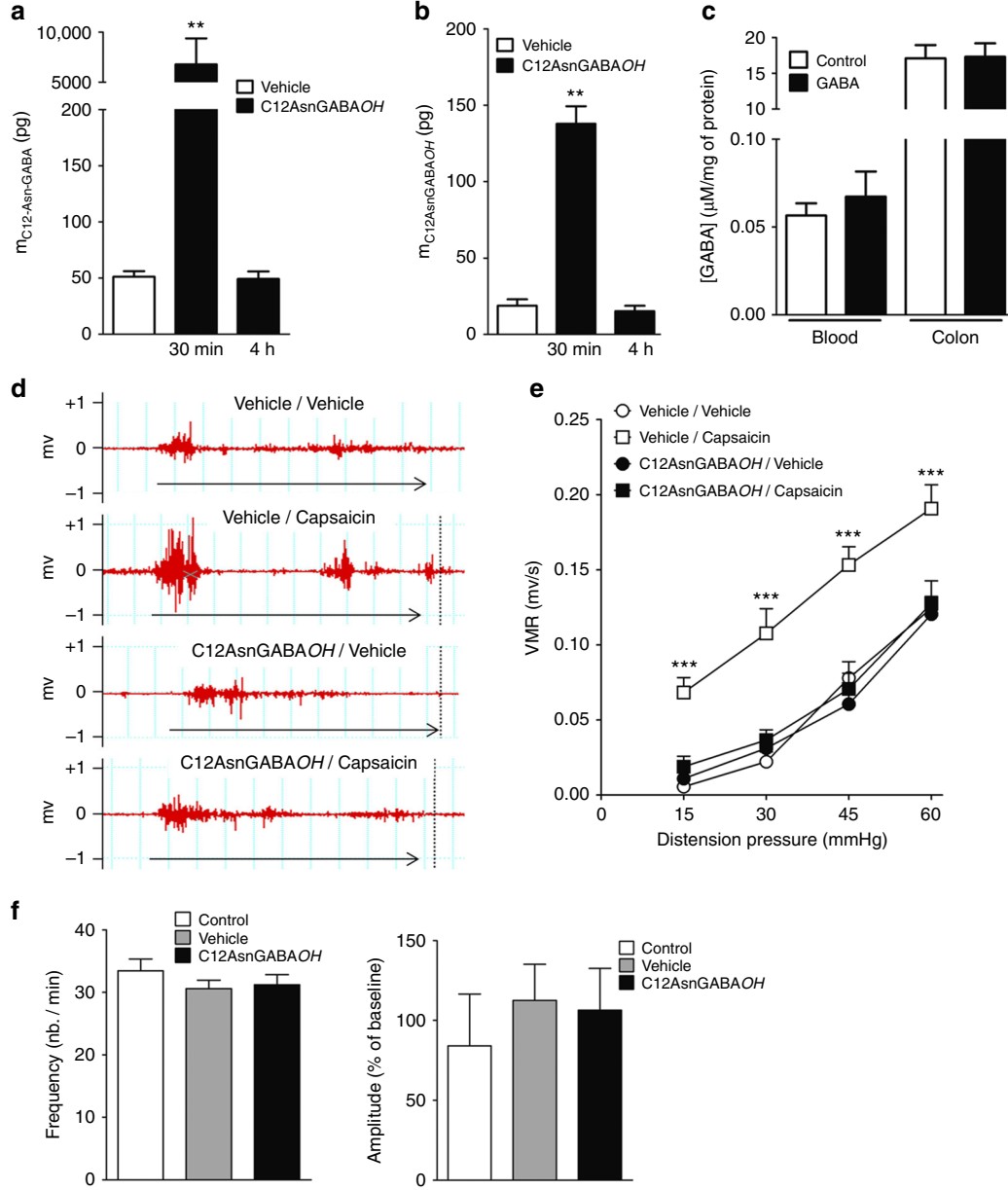

**Fig. 7** C12AsnGABA inhibits capsaicin-induced VH without altering intestinal contraction. Mice received intracolonic administration of C12AsnGABA*OH* (10 μM; black bars) or vehicle (40% ethanol; white bars) and 30 min or 4 h later colon (**a**) and blood (**b**) were harvested in order to quantify C12AsnGABA*OH* by LC-MS/MS. Data are expressed as mean ± SEM, $n = 8$ mice per group. Statistical analysis was performed using Kruskal–Wallis analysis of variance and subsequent Dunn's post hoc test. ***$p < 0.001$ significantly different from vehicle group. **c** Mice received intracolonic administration of GABA (10 μM; black bars) or vehicle (white bars) and 30 min colon and blood were harvested in order to quantify GABA. Data are expressed as mean ± SEM, $n = 5$ mice per group. Statistical analysis was performed using Kruskal–Wallis analysis of variance and subsequent Dunn's post hoc test. **d** Representative traces of mouse abdominal muscle contractions in response to 60 mm Hg CRD, in mice pretreated with C12AsnGABA*OH* (10 μM) or vehicle and treated with capsaicin (100 μg per mouse) or its vehicle. The arrows represent the time of distension (10 s). **e** Mice received intracolonic administration of C12AsnGABA*OH* (10 μM; black symbols) or vehicle (40% ethanol; white symbols) and 30 min or 4 h later an intracolonic administration of capsaicin (100 μg per mouse; square) or its vehicle (40% ethanol; circle). Fifteen minutes after capsaicin or vehicle treatment, VMR to increasing pressures of CRD was performed. Data are expressed as mean ± SEM, $n = 7$–8 mice per group. Statistical analysis was performed using two-way Anova analysis of variance and subsequent Bonferroni post hoc test. ***$p < 0.001$ significantly different from vehicle/vehicle group. Ex vivo measurement of duodenal mechanical contraction frequency (**f**, left panel) and amplitude (**f**, right panel) in response to Krebs–Ringer solution (control; white bar), DMSO 0.2% (vehicle; gray bar), or to C12AsnGABA*OH* (10 μM; black bar). Data are expressed as mean ± SEM, $n = 5$–6 per group. Statistical analysis was performed using Kruskal–Wallis analysis of variance and subsequent Dunn's post hoc test

**C12AsnGABA*OH* crosses the epithelial barrier.** The initial barrier for any drug absorption is the intestinal epithelial cell wall after penetrance of the mucus layer. To evaluate whether C12AsnGABA*OH* is capable of crossing the epithelial barrier and

then stimulate GABA receptors on neurons[29], human epithelial cells monolayers (fully differentiated Caco-2 cells) were treated at the apical side with C12AsnGABA*OH*. LC-MS/MS quantification of this compound was performed in cells and in apical and

basolateral side of transwell chambers after an incubation period of 24 h. Approximately 50% of C12AsnGABAOH added in the apical chamber (800 ng) was found in the basolateral chamber (Fig. 6a), whereas epithelial cells contained low levels of C12AsnGABAOH after 24 h. The transport of GABA alone across the cell monolayer was also assessed. For this purpose, commercial GABA was added in the apical chamber (800 ng) and after 3, 6, and 16 h, the presence of this molecule was quantified by LC-HRMS in basal and apical chambers (Fig. 6b). GABA was not detected in the basal chamber following the incubation period, showing that this molecule did not cross the intestinal epithelial monolayer. Thus, the addition of the (C12AsnOH) by the bacteria to the GABA confers the capacity for this neuromediator to cross the intestinal epithelial barrier.

To assess the effect of C12AsnGABAOH on paracellular permeability, transport of dextran 4 kDa fluorescein isothiocyanate (FITC) across a Caco-2 cell monolayer was investigated. As shown in the Fig. 5c, the percentage of 4 kDa FITC traversing the cell monolayers after 24 h was not modified by C12AsnGABAOH (10 μM) treatment, evidencing that C12AsnGABAOH does not alter paracellular permeability. In parallel, the release of CXCL8 from Caco-2 cells into the medium, of both apical and basolateral sides of the transwell, was assessed by ELISA. None of the tested doses (1, 10, 100 μM) and neither the vehicle modified the secretion of CXCL8 by Caco2 cells (Fig. 6d). Thus, the addition of (C12AsnOH) to GABA by the bacteria confers the capacity for this neuromediator to cross the intestinal epithelial barrier without altering paracellular permeability. We hypothesized that intracolonic administration of C12AsnGABAOH could mimic the luminal production of the analgesic lipopeptide by EcN and its diffusion across the epithelial barrier.

**C12AsnGABAOH inhibits VH.** In a first set of experiments, we assessed the in vivo ability of C12AsnGABAOH and GABA alone to cross the epithelial barrier. Intracolonic administration of C12AsnGABAOH in mouse increased the concentration of this compound in the colonic tissue and blood (Fig. 7a, b). In contrast, following its intracolonic administration, GABA concentration were not increased in the colonic wall or in the blood (Fig. 7c). Based on the inhibitory effect of C12AsnGABAOH on calcium mobilization in sensory neurons and on its capacity to cross the epithelial barrier in vivo, we evaluated the analgesic potency of this lipoamino acid. The impact of C12AsnGABAOH on VH was assessed by measuring visceromotor responses (VMR) to colorectal distension (CRD). VMR recordings were initiated 15 min after intracolonic administration of either capsaicin (100 μg per animal; 100 μL) or the vehicle (EtOH 40%). Capsaicin evoked an increase ($p < 0.05$) in VMR to CRD pressures of 15–60 mm Hg compared to vehicle (Fig. 7d, e). This increase was significantly prevented for all distension pressures in animals pretreated with a 100 μL intracolonic injection of C12AsnGABAOH (10 μM), which showed similar VMR values to those obtained in the control group (vehicle/vehicle). C12AsnGABAOH administration in the absence of capsaicin did not modify visceral sensitivity in response to CRD (Fig. 7d, e). IBS patients are characterized by diverse alterations of gut transit: diarrhea, constipation, or mixed[1], thus a drug impairing bowel motility could limit its putative use in this pathology. We evaluated whether C12AsnGABAOH was capable of impairing duodenal motility. For this purpose, isotonic sensors were used to measure ex vivo mechanical contractions. Application of C12AsnGABAOH on ex vivo duodenal preparations impaired neither the frequency of contractions (Fig. 7f), nor the amplitude of intestinal contractions (Fig. 7f) compared to control and vehicle.

## Conclusion

Here, we identified a lipopeptide related to GABA—the main inhibitory transmitter of the central nervous system—exhibiting analgesic properties in visceral pain. Three enzymes implicated in the synthesis of this C12AsnGABAOH have been identified. Surprisingly, these enzymes were encoded by a genomic island, named *pks*, which carries the cluster of genes that enables the synthesis of hybrid peptide polyketides and especially the genotoxin colibactin, a bona fide virulence factor and a putative carcinogenic agent. Our results illustrate how the colibactin NRPS–PKS biosynthetic pathway represents a rich source of unusual assembly-line enzymology coding for additional bioactive compounds distinct from colibactin. The addition of the C12AsnOH moiety confers to GABA the capacity to diffuse across the epithelial barrier and subsequently to act on sensory neurons. Interestingly, C12AsnGABAOH does not modify the physiology of the intestinal epithelium or the intestinal motility, suggesting that it might have fewer side effects than prototypical analgesics such as morphine. Thus, C12AsnGABAOH may represent a promising therapeutic agent for the management of visceral pain.

## Methods

**Chemicals.** Lipoxin A4 deuterated (LxA4-d5); leukotriene B4 deuterated (LTB4-d4) and 5S-hydroxy-eicosatetraenoic acid deuterated (5-HETE-d8) were purchased from Cayman Chemicals (Interchim, Montluçon, France). Methanol (MeOH), Hank's balanced salt solution (HBSS), HEPES, Collagenase type I from Clostridium histolyticum, dispase II, papain, cytosine β-D arabinofuranoside (ARAc), 5-fluorouracil (5-FU), uridine, 4 kilodalton FITC-labeled dextran, capsaicin (transient receptor potential vanilloid [TRPV1]-1 agonist), pluronic F-127 were obtained from Sigma-Aldrich (Saint Quentin Fallavier, France). Saclofen (GABA$_B$ receptor antagonist) and GABA were obtained from Tocris Bioscience (R&D Systems Europe, Lille, France). The mix of GPCR agonists (histamine, bradykinin, serotonin) was provided by Sigma-Aldrich (Saint Quentin Fallavier, France), Fluo-4 acetoxymethyl (Fluo4 AM, molecular probes) was from Life Technologies (Eugene, OR, USA), Leibovitz's L-15 Medium was from Gibco (Invitrogen Life Technologies, Paisley, UK).

**Animals.** Male C57Bl6 mice (6–8 weeks, Janvier St Quentin-Fallavier, France) were used to produce primary cultures of DRG sensory neurons for calcium flux experiments, and to perform studies of colorectal distention and intestinal isotonic contraction. All procedures were performed in accordance with the Guide for the Care and Use of Laboratory Animals of the European Council and were approved by the Animal Care and Ethics Committee of US006/CREFE (CEEA-122).

**Bacterial strains and culture conditions.** *E. coli* MG1655, MG1655+BAC pks+, EcN 1917 (Mutaflor, DSM 6601, serotype O6:K5:H1), the isogenic mutants EcNΔclbA, EcNΔclbN, EcNΔclbB, EcNΔclbC, and EcNΔclbP were used in this study (Supplementary Table 2). Gene inactivation was engineered using the lambda red recombinase method[30] and deletions were confirmed by using flanking primers. MG1655, MG1655+BAC pks+, EcNwt, EcNΔclbA, EcNΔclbN, EcNΔα, EcNΔclbC, and EcNΔclbP were grown on LB agar plates supplemented with kanamycin (50 μg mL$^{-1}$) or chloramphenicol (25 μg mL$^{-1}$) when required. After overnight incubation at 37 °C, single colonies for each strain were seeded in 4 mL of LB with antibiotics (when needed) and incubated overnight at 37 °C in agitation (250 rpm). Bacteria were then inoculated at $OD_{600} = 0.1$ in 10 mL of minimal medium A (K$_2$HPO$_4$ 10.5 g L$^{-1}$; KH$_2$PO$_4$ 4.5 g L$^{-1}$; (NH$_4$)$_2$SO$_4$ 1 g L$^{-1}$; sodium citrate 0.5 g L$^{-1}$; MgSO$_4$ 0.2 g L$^{-1}$, and glycerol 0.2%) supplemented with kanamycin or chloramphenicol (when required) and cultures were grown for 24 h at 37 °C under shaking conditions (250 rpm). The last step was repeated one more time.

**Extraction of lipoamino acids and lipopeptides.** Bacterial cultures were centrifuged at 755 g for 15 min and the recovered bacterial pellets were immediately crushed with a FastPrep-24 Instrument (MP Biomedical, Santa Ana, CA) in 200 μL HBSS and 5 μL internal standard (IS) mixture (Deuterium-labeled compounds) (400 ng mL$^{-1}$). After two crush cycles (6.5 m s$^{-1}$, 30 s), 10 μL of suspensions were withdrawn for protein quantification and 0.3 mL of cold methanol (MeOH) was added. In parallel, bacteria culture supernatants were filtered through a 0.22-μm pore size filter (Millipore) to remove residual bacterial cells, and 1 mL of supernatant was collected for lipid extraction after addition of 5 μL IS mixture and 0.3 mL of cold MeOH. Both bacteria and supernatant samples were centrifuged at $1016 \times g$ for 15 min (4 °C) and the resulting supernatants were submitted to solid-phase extraction of lipids using HRX-50 mg 96-well plates (Macherey Nagel,

Hoerd, France). Briefly, plates were conditioned with 2 mL MeOH and 2 mL H₂O/ MeOH (90:10, v/v). Samples were loaded at a flow rate of about one drop per 2 s and, after complete loading, columns were washed with 2 mL H₂O/MeOH (90:10, v/v). The columns were thereafter dried under aspiration and lipids were eluted with 2 mL MeOH. Solvent was evaporated under N₂ and samples were resuspended in 10 μL MeOH for liquid chromatography/tandem mass spectrometry analysis[31].

**Characterization of C12-Asn-aminobutyric acid.** The characterization of C12-Asn-aminobutyric acid was performed on a high-performance liquid chromato-graph (U3000, Thermo Fisher Scientific, Waltham, MA, USA) coupled on line to a Fourier Transform Mass Spectrometer (FT/MS) (QExactive+ high-resolution mass spectrometer, Thermo Fisher Scientific, Bremen, Deutschland). MS and MS/MS analyses were performed in the negative ion FTMS mode at a resolution of 30,000 (at $m/z$ 400) with the following source parameters: capillary temperature was 325 ° C, source heater temperature was 300 °C, sheath gas flow rate was 30 a.u. (arbitrary unit), auxiliary gas flow rate was 10 (a.u.), and source voltage was −2.9 kV. Samples were injected on a ZorBAX SB 120 C18 column (2.1 × 100 mm, 2.7 μm) (Agilent Technologies) maintained at 40 °C. Solvent A was 0.1% formic acid in H₂O and solvent B was 0.1% formic acid in acetonitrile at a flow rate of 350 μL min⁻¹. The multi step gradient was as follows: 25% B at 0 min, 40% B at 10 min, 47.5% B at 10.1 min, 73.8% B at 21 min, 100% B at 21.2 min, 100% B at 23.2 min, 25% B at 23.4 min, and 25% B at 25.4 min. The autosampler was set at 5 °C and the injection volume was 5 μL. The identification was performed using XCalibur software (Thermo Fisher Scientific). The MS/MS experiments were performed by using collisional activation within non-resonant excitation mode (HCD[32]). The excita-tion energy was optimized in the range of the 20 and 35% NCE[33] such as the precursor ion survived to the dissociation processes within a relative abundance between 30 and 150% of the most abundant product ion displayed in the HCD spectrum recorded with a resolution of 30,000.

**Synthesis of C12AsnAABA*OH*, C12AsnBBA*OH*, and C12AsnGABA*OH*.** All reactions requiring anhydrous conditions were conducted in flame dried glassware with magnetic stirring under an atmosphere of nitrogen unless otherwise men-tioned. Anhydrous CH₂Cl₂ was obtained from the Innovative Technology PS-Micro solvent purification system. Other solvents and reagents were used as obtained from the suppliers (Aldrich, Alfa Aesar, Acros) unless otherwise noted. Reactions were monitored by TLC using plates precoated with silica gel 60 (Merck). Reaction components were visualized by using a 254 nm UV lamp and treatment with basic KMnO₄ solution. Column chromatography was performed by using silica gel 40–63 μm. ES-MS data were obtained using the mass spectrometers LCQ Fleet (Thermo Fisher Scientific) with the following source parameters: capillary temperature was 300 °C, sheath gas flow rate was 25, and the source voltage was 5 kV. High-resolution mass data were acquired at Laboratoire de Mesures Physi-ques (Université de Montpellier) on a Synapt G2-S from Waters equipped with an ESI source used in high-resolution positive mode. Data were obtained in a window between $m/z$ 50 and $m/z$ 1500 with a capillary tension of 3 kV, a cone voltage of 30 V, capillary temperature was 400 °C, and source block temperature was 140 °C. ¹H NMR spectra (Supplementary Figs. 6a, 8a, 17a) were obtained at 300 or 500 MHz on Bruker spectrometers. The spectra were recorded in MeOD. The ¹H NMR spectra are reported as follow: chemical shift in ppm [multiplicity, coupling constant(s) $J$ in Hz, relative integral]. The multiplicities are defined as follow: br. = broad, $m$ = multiplet, $s$ = singlet, $d$ = doublet, $t$ = triplet, q = quadruplet, quint. = quintuplet, or combinations thereof. ¹³C NMR spectra (Supplementary Figs. 6b, 7b, 8b) were recorded in MeOD. LCMS analyses were carried out on a Waters Micromass with Alliance 2695 chain with a Chromolite HR C18 column (25 × 4.6 mm, Merck Inc.) monitoring at 214 nm with positive ion mode for the ion $m/z$ value detection. Solvents for LCMS were water with 0.1% formic acid (solvent A) and acetonitrile with 0.1% formic acid (solvent B). Compounds were eluted at a flow rate of 3 mL min⁻¹ by a linear gradient of 0–100% solvent B over 2.5 min, and finally 100% solvent B for 1 min before equilibrating the column back to 0% solvent B over 1 min.

*LC-MS purification*: Samples were prepared in DMSO. The LC/MS autopurification system consisted of a binary pump Waters 2525, an injector/ fraction collector Waters 2676, coupled to a Waters Micromass ZQ spectrometer (electrospray ionization mode, ESI+). Purifications were carried out using a Luna® 5 μm C18 100 Å, LC Column 100 × 21.2 mm, AXIA™ Packed. A flow rate of 20 mL min⁻¹ and a gradient of 40–60% B over 10 min were used. Eluent A: water with 0.1% TFA; eluent B: acetonitrile with 0.1% TFA. Positive ion electrospray mass spectra were acquired at a solvent flow rate of 204 μL min⁻¹. Nitrogen was used for both the nebulizing and drying gas. The mass spectrum data were obtained in a scan mode ranging from 100 to 1000 $m/z$ in 0.1 s intervals; 10 scans were summed up to get the final mass spectrum. Collection control trigger is set on single protonated ion with a MIT (minimum intensity threshold) of 7.105.

As illustrated in Supplementary Fig. 5, syntheses were performed following seven different steps:

(1)  CBz-GABA-OH: A solution of (S)-γ-aminobutyric acid (5.22 g, 50.6 mmol, 1 eq.) in 2 M NaOH solution (25 mL, 50 mmol, 1 eq.) was cooled to 0 °C and treated with benzyl chloroformate (8.23 mL, 55.6 mmol, 1.15 eq.), while pH is maintained around 10 by continuous addition of 3 M NaOH solution.

After 15 min, the reaction was allowed to stir at room temperature for 3 h. After two extractions with Et₂O, the pH of the aqueous solution was adjusted to 1.5 by addition of 6 M HCl solution. After having saturated with solid NaCl, the aqueous layer was extracted with EtOAc (3 × 50 mL). The combined organic layers were washed with brine (3 × 25 mL), dried over MgSO₄, and evaporated under vacuum. The oily residue was taken up with Et₂O and the solvent removed again to give a white solid of CBz-GABA-OH (11.3 g) used directly in the next step.

(2)  CBz-GABA-OtBu: To a solution of CBz-GABA-OH (11.3 g), tBuOH (14.5 mL, 152 mmol, 3 eq.), and DMAP (620 mg, 5.1 mmol, 0.1 eq.) in CH₂Cl₂, after cooling at 0 °C, was added dicyclohexylcarbodiimide (12.52 g, 60.7 mmol, 1.2 eq.). After 1 h, the reaction was stirred vigorously overnight at room temperature. DCU was filtered and washed with EtOAc (2 × 5 mL). The filtrate was washed with 1 M HCl solution (50 mL), and refiltrated. The aqueous layer was extracted with CH₂Cl₂ (3 × 50 mL). The combined organic layers were washed with brine (50 mL), 5% NaHSO₄ solution (50 mL), and brine (50 mL) again. The layer was dried over MgSO₄ and evaporated under vacuum. The crude was purified by column chromatography (pentane/ AcOEt 90/10 to 70/30) to obtain CBz-GABA-OtBu (7.39 g, 50%). MS (ESI+) [M+Na]⁺: 316.17; [2M+Na]⁺: 609.00; MS (ESI−) [M]⁻: 293.33.

(3)  GABA-OtBu.HCl: A solution of CBz-GABA-OtBu (2 g, 6.82 mmol, 1 eq.) in MeOH (20 mL) was treated with Pd/C (10%, 200 mg, 10% w/w). After 6 h, the catalyst was filtered over a celite® pad. The solvent was carefully evaporated (caution, final product is volatile). 1 M HCl solution (50 mL) was added. The aqueous layer was extracted with EtOAc (2 × 50 mL) and pH was adjusted to 10 by addition of NaOH pellets. The aqueous layer was extracted with EtOAc (3 × 50 mL). This last organic phase was washed with brine (2 × 25 mL), dried over MgSO₄, and carefully evaporated. The crude reaction mixture was dissolved in Et₂O (25 mL) and pH was adjusted to 1 by addition of 2 M HCl solution in Et₂O. After 30 min of stirring, 676 mg GABA-OtBu. HCl (51% yield) was filtered and dried.

(4)  Fmoc-(S)Asn(Trt)-GABA-OtBu: To a solution of Fmoc-(S)Asn(Trt)-OH (328 mg, 0.55 mmol, 1.1 eq.) and GABA-OtBu.HCl (98 mg, 0.5 mmol, 1 eq.) in CH₂Cl₂ (4 mL) was added HBTU (176 mg, 0.75 mmol, 1.5 eq.), HOBt (7 mg, 0.05 mmol, 0.1 eq.), and N-methyl morpholine (176 μL, 1.6 mmol, 3.2 eq.). The mixture was stirred overnight. The solvent was evaporated in presence of silica gel. The product was purified by column chromatography (CH₂Cl₂/MeOH 97.5/2.5) to obtain Fmoc-Asn(Trt)-GABA-OtBu (360 mg, 97%). MS (ESI+) [M+H]⁺: 738.17; [M+Na]ff: 760.50; MS (ESI−) [M-Fmoc]⁻: 514.58.

(5)  Asn(Trt)-GABA-OtBu: To a solution of Fmoc-Asn(Trt)-GABA-OtBu (350 mg, 0.47 mmol, 1 eq.) in CH₂Cl₂ (1.35 mL) was added diethylamine (1.35 mL) and stirred for 2 h at room temperature. The resulting solution was concentrated in vacuum and used directly in the amine coupling step.

(6)  C12:0-Asn(Trt)-GABA-OtBu: To a solution of Asn(Trt)-GABA-OtBu (0.47 mmol, 1 eq.) and lauric acid (143 mg, 0.71 mmol, 1.5 eq.) in CH₂Cl₂ (4 mL) was added HBTU (194 mg, 0.83 mmol, 1.75 eq.), HOBt (6 mg, 0.04 mmol, 0.1 eq.), and N-methyl morpholine (195 μL, 1.77 mmol, 3.75 eq.). The mixture was stirred overnight. The solvent was evaporated in presence of silica gel. The product was purified by column chromatography (pentane/ AcOEt 75/25 to 60/40) to obtain C12:0-Asn(Trt)-GABA-OtBu (235 mg, 71%). MS (ESI+) [M+H]⁺: 698.25; [M+Na]⁺: 720.50; MS (ESI−) [M]⁻: 696.75.

(7)  C12:0-Asn-GABA-OH: To a solution of C12:0-Asn(Trt)-GABA-OtBu (235 mg, 0.33 mmol, 1 eq.) in CH₂Cl₂ (0.5 mL) was added TFA (0.5 mL) and stirred for 3 h at room temperature. The resulting solution was concentrated in a vacuum. Traces of TFA were eliminated by coevaporation with acetonitrile (3 × 2 mL). The crude reaction mixture was stirred with diisopropyl ether (10 mL) and filtrated. The resulting white solid (102 mg) was purified by HPLC to give 11 mg (8.6%) of C12:0-Asn-GABA-OH. Analytical LC-MS Rt: 1.62 mn;

$[\alpha]_D^{25}$ = −16.5 (c 2, DMSO); ¹H NMR (300 MHz, DMSO; Supplementary Fig. 12a) δ 7.87 (d, $J$ = 7.9 Hz, 1H), 7.72 (t, $J$ = 5.4 Hz, 1H), 7.22 (s, 1H), 6.81 (s, 1H), 4.43 (q, 1H), 2.99 (q, $J$ = 6.2 Hz, 3H), 2.41 (dd, $J$ = 15.2, 6.0 Hz, 1H), 2.28 (dd, $J$ = 15.2, 7.6 Hz, 1H), 2.13 (t, $J$ = 7.3 Hz, 2H), 2.04 (t, $J$ = 7.3 Hz, 2H), 1.66–1.48 (m, 2H), 1.41 (s, 2H), 1.19 (sl, 18H), 0.81 (t, $J$ = 6.3 Hz, 3H). ¹³C NMR (75 MHz, DMSO; Supplementary Fig. 12b) δ 174.82, 172.51, 171.90, 171.48, 50.15, 38.44, 37.82, 35.68, 31.77, 31.45, 29.50, 29.49, 29.42, 29.31, 29.19, 29.10, 25.61, 24.97, 22.57, 14.43. HRMS [M+H]⁺ calc: 400.2811 found: 400.2804.

C12:0-Asn-AABA-OH was synthesized following the procedure described above using (S)-α-aminobutyric acid as a starting material. At the last step of the synthesis, 300 mg of C12:0-Asn(Trt)-AABA-OtBu was deprotected and purified by HPLC to give C12:0-Asn-AABA-OH (17 mg, 10%). In this case, epimerisation of α-amino butyric acid moiety was observed during the synthesis. Analytical LC-MS Rt: 1.65, 1.69 mn;

$[\alpha]_D^{25}$ = −11.0 (c 2, DMSO); ¹H NMR (300 MHz, DMSO; Supplementary Fig. 10a) δ 7.97 (t, $J$ = 8.0 Hz, 1H), 7.80 (d, $J$ = 7.8 Hz, 1H), 7.75 (d, $J$ = 7.9 Hz, 0H), 7.25 (s, 1H), 6.85 (s, 1H), 4.63–4.49 (m, 1H), 4.16–3.99 (m, 1H), 2.52–2.40 (m, 1H, determined with COSY experiment), 2.29 (dd, $J$ = 15.4, 8.2 Hz, 1H), 2.04 (t, $J$ = 7.3 Hz, 2H), 1.79–1.49 (m, 2H), 1.49–1.32 (m, 2H), 1.19 (sl, 16H), 0.88–0.72 (m, 6H). ¹³C NMR (75 MHz, DMSO; Supplementary Fig. 10b) δ 173.70, 173.63, 172.77,

172.72, 171.85, 171.77, 171.65, 171.47, 53.55, 53.47, 49.93, 49.81, 37.92, 37.45, 35.65, 31.77, 29.49, 29.40, 29.32, 29.19, 29.08, 25.67, 24.95, 24.81, 22.57, 14.43, 10.35, 10.26. HRMS [M+H]$^+$calc: 400.2811 found: 400.2814.

C12:0-Asn-BABA-OH was synthesized following the procedure described above using (S)-α-aminobutyric acid as a starting material. At the last step, 280 mg of C12:0-Asn(Trt)-BABA-OtBu was deprotected and purified by HPLC to give C12:0-Asn-BABA-OH (21 mg, 13%). Analytical LC-MS Rt: 1.16 mn;

[α]$_D^{25}$ = −6.0 (c 5, DMSO); $^1$H NMR (300 MHz, DMSO; Supplementary Fig. 11a) δ 7.88 (d, $J$ = 7.9 Hz, 1H), 7.68 (d, $J$ = 8.0 Hz, 1H), 7.20 (s, 1H), 6.81 (s, 1H), 4.42 (q, $J$ = 7.2 Hz, 1H), 4.07–3.86 (m, 1H), 2.51–2.11 (m, 5H), 2.03 (t, $J$ = 7.1 Hz, 2H), 1.41 (s, 2H), 1.19 (sl, 16H), 1.01 (d, $J$ = 6.4 Hz, 3H), 0.80 (t, $J$ = 6.5 Hz, 3H). $^{13}$C NMR (75 MHz, DMSO; Supplementary Fig. 11b) δ 173.02, 172.52, 171.84, 170.67, 50.08, 42.16, 40.90, 37.95, 35.64, 31.77, 29.50 (2C), 29.43, 29.32, 29.19, 29.07, 25.65, 22.57, 20.34, 14.42. HRMS [M+H]$^+$calc: 400.2811 found: 400.2815.

**Quantification of C12AsnGABA*OH* and C12AsnBABA*OH***. The quantification of C12AsnGABA*OH* and C12AsnBABA*OH* was performed on a high-performance liquid chromatography (Agilent 1290 Infinity) coupled to a triple quadrupole mass spectrometer (G6460 Agilent). Samples were injected on a ZorBAX SB 120 C18 column (2.1 × 100 mm, 2.7 μm) (Agilent Technologies) maintained at 40 °C. Solvent A was 0.1% formic acid in H$_2$O and solvent B was 0.1% formic acid in acetonitrile at a flow rate of 350 μL min$^{-1}$. The linear gradient was as follows: 30% B at 0 min, 85% B at 15 min, 100% B at 15.1 min, 100% B at 16.5 min, and 30% B at 16.7 min. The autosampler was set at 5 °C and the injection volume was 5 μL. The HPLC system was coupled online to an Agilent 6460 triple quadrupole MS (Agilent Technologies) equipped with an ESI source. ESI was performed in negative ion mode. Source parameters used were as follows: source temperature was set at 325 ° C, nebulizer gas (nitrogen) flow rate was 10 L min$^{-1}$, sheath gas temperature was 400 °C, sheath gas (nitrogen) flow rate was 12 L min$^{-1}$, and the spray voltage was adjusted to −3.5 kV. Analyses were performed in Selected Reaction Monitoring detection mode using nitrogen as collision gas. The specific transition was {$m/z$ 398}/{$m/z$ 295} corresponding to the [M-H]$^-$/[(M-H)ABA]$^-$ abundance ratio (product ions displayed in Fig. 2a). Fragmentor and collision energy were, respectively, fixed at 80 and 18 V. Peak detection, integration and quantitative analysis were done using Mass Hunter Quantitative analysis software (Agilent Technologies). Finally, the quantification of C12AsnGABA*OH* and C12AsnBABA*OH* were performed using a calibration curve calculated by the IS method. Six biological replicates were performed for each strain.

**GABA analysis by LC-HRMS**. GABA was analyzed by liquid chromatography (Vanquish System, Dionex, Sunnyvale, CA, USA) coupled to a Orbitrap Q Exactive plus mass spectrometer (Thermo Fisher Scientific, Waltham, MA, USA) equipped with a heated ESI probe. MS analyses were performed in the positive FTMS mode at a resolution of 70,000 (at $m/z$ 400) with the following source parameters: capillary temperature was 320 °C, source heater temperature was 250 ° C, sheath gas flow rate was 40, auxiliary gas flow rate was 15, S-Lens RF level was 50%, and source voltage was 4 kV. Samples were injected on a Supelco HS F5 Discovery column (150 mm × 2.1 mm; 5 μm particle size) equipped with a Supel-coHSF5 guard column (20 mm × 2.1 mm; 5 μm particle size). Solvent A was 0.1% formic acid in H$_2$O and solvent B was 0.1% formic acid in acetonitrile at a flow rate of 250 μL min$^{-1}$. The solvent B was varied as follows: 0 min: 2%, 2 min: 2%, 5 min: 5%, 5.1 min: 100%, 9 min: 100%, 9.1 min: 2%, and 14 min: 2%. The volume of the injection was 5 μL. Identification was determined by extracting the accurate mass of GABA at $m/z$ 104.0702 with a mass accuracy of 5 ppm. The IS used corresponds to U-$^{13}$C-GABA and was extracted at 108.0840 $m/z$. For each sample, 50 μL of IS was added in solution of extraction used for blood and colon samples. Samples of cell supernatants were mixed with the IS at a 1:1 ratio. The retention time and the accurate mass of the compound were validated with a commercial standard of GABA purchased from Sigma-Aldrich (St. Louis, MO, USA). Data treatment was done with TraceFinder™ Software (Thermo Fisher Scientific). The limit of quantification (LOQ) is the lowest concentration of analyte in a sample which can be quantified reliably. The LOQ is considered being the lowest calibration standard with an accuracy and precision less than 20%. The limit of detection (LOD) is the lowest concentration of an analyte that the can reliably differentiate from background noise. It corresponds to three times the background noise. For the GABA, under our condition, the LOD was 0.43 μM and the LOQ was 1.73 μM.

**Cell culture and absorption/permeability experiments**. Caco-2 cells (Leibniz Institute DSMZ—German Collection of Microorganisms and Cell Cultures GmbH, catalog number: ACC 169) were grown in Glutamax DMEM (Gibco, Invitrogen Life Technologies, Paisley, UK) supplemented with 10% heat-inactivated fetal bovine serum (Gibco, Invitrogen Life Technologies, Paisley, UK), 1% nonessential amino acids, and 1% antibiotics (100 U ml$^{-1}$ penicillin and 100 mg ml$^{-1}$ streptomycin; Gibco, Invitrogen Life Technologies, Grand Island, NY, USA) at 37 °C in a 5% CO$_2$ water-saturated atmosphere. Cells were seeded on Transwell inserts (Costar, Sigma-Aldrich, Saint-Quentin Fallavier, France) and absorption studies of C12AsnGABA*OH* and GABA were performed 16 days later. These compounds were quantified by LC-MS/MS in both apical and basolateral chambers as well as in cells. The effect of C12AsnGABA*OH* (1–100 μM) on paracellular permeability was also measured by mucosal-to-serosal flux of 4 kDa FITC-labeled dextran as

previously described[34]. All experiments were performed in serum-free medium. CXCL8 concentration in apical and basolateral chambers was determined 24 h after C12AsnGABA*OH* addition using a commercial enzyme-linked immunosorbent assay kit (BD Biosciences, Erembodegen, Belgium) following manufacturer's recommendations[35].

**Calcium imaging of sensory neurons**. Mouse DRG were dissociated as previously described[36]. After 40–42 h of culture, cells were incubated with HBSS containing 20 mM HEPES, 1 mM fluo-4 acetoxymethyl (AM), and 20% pluronic F-127 for 30 min at 37 °C plus 30 min in the dark at RT. The plates were then washed with HBSS and 100 μl of HBSS were added to each well. Live cell imaging of calcium was carried out on an automated microscope (Apotome 2, Zeiss) with a ×10 objective. Images were acquired using the Zen imaging software and a kinetic of 80 recordings (one per second) was performed. The first five images were used to determine the baseline and, from 6 to 60 s, neurons were exposed to either a mix of GPCR agonists (histamine, bradykinin, serotonin, 10 μM), capsaicin (125 nM), or vehicle (HBSS). After 60 s, neurons were treated with KCl (50 mM) in order to discriminate neurons from glial cells. The ImageJ software was used to perform the analysis of calcium flux. In a first set of experiments, neurons were pre-treated with either C12-Asn-GABA, C12-Asn-BABA, or C12-Asn-AABA (0.1, 1, and 10 μM), or vehicle (HBSS/DMSO 0.01%) for 5 min and then were stimulated with either capsaicin or the mix of GPCR agonists. In a second set of experiments, neurons were pre-incubated for 5 min with saclofen (10, 50, and 100 μM) or its vehicle (HBSS/DMSO 0.01%), then treated for 5 min with C12-Asn-GABA (10 μM) or vehicle, and finally stimulated with either capsaicin or the mix of GCPR agonists and vehicle.

**CRD and electromyography (EMG) recordings**. Nickel-chrome electrodes were implanted in the abdominal external oblique musculature of anesthetized mice in order to detect EMG activity as previously described[37]. CRD was performed 3 days post-surgery by inserting a distension catheter (Fogarty catheter for arterial embolectomy, 4 F; Edwards Lifesciences, Nijmegen, the Netherlands) into the colon at 5 mm from the anus. The balloon was progressively inflated in a stepwise of 15 mm Hg (from 0 to 60 mm Hg) performing 10-s distension for each pressure (in triplicate) and with resting intervals of 5 min as previously described[37]. In a first set of experiments, four groups of mice ($n$ = 7–8 per group) were pre-treated with a 100 μL intracolonic injection of either C12-Asn-GABA (10 μM) in EtOH 40% or vehicle, and 30 min later animals were administered either capsaicin (100 μg per animal) or vehicle (EtOH 40%) intracolonically. CRD was performed 15 min after capsaicin/vehicle administration, and VMR to different colorectal pressures were recorded.

**Intestinal isotonic contractions**. Duodenum segments obtained from euthanized mice were washed in Krebs–Ringer bicarbonate/glucose buffer (pH 7.4) in an atmosphere of 95% O$_2$–5% CO$_2$. Duodenum segments were then incubated in oxygenated Krebs–Ringer solution and attached to an isotonic transducer as previously described[38]. Isotonic contractions were recorded by means of BDAS software (Hugo Sachs Elektronik) following the transducer displacement. Basal contractions were recorded after duodenum segment attachment for 10 min. Subsequently, 100 μL of either C12-Asn-GABA (10 μM) or Krebs–Ringer solution were added in survival medium, and contractions were recorded for 10 min. The amplitudes were also recorded for 10 min at 10-s intervals, and their average was compared to average basal contractions. Contractions amplitudes are presented as percentage relative to basal response, while contraction frequencies are expressed as a number of contractions per min.

**Statistical analysis**. Data are presented as means ± SEM. The software GraphPad Prism 6.0 (GraphPad, San Diego, CA) was used for statistical analysis. Multiple comparisons within groups were performed by Kruskal–Wallis test followed by Dunns post-test or by two-way Anova followed by Bonferroni post-test. Statistical significance was accepted at $p < 0.05$.

**Data availability**. The authors declare that the relevant data supporting the findings of the study are available in this article and its Supplementary Information, or from the corresponding author upon request.

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

## Acknowledgements

We gratefully acknowledge the microscope core facility, INSERM UMR1043, Toulouse, the animal care facility, Genetoul, anexplo, US006/INSERM, Toulouse, and the Toulouse INSERM Metatoul-Lipidomique Core Facility-MetaboHub ANR-11-INBS-010, where lipidomic analysis were performed. This work was supported by the Région Midi-Pyrénées (to Nicolas Cenac). This work was also supported by research grant ANR-13-BSV1-0028-01 from the Agence Nationale de la Recherche and the platform Anin-fimip, an EquipEx ('Equipement d'Excellence') supported by the French government through the Investments for the Future program (ANR-11-EQPX-0003).

## Author contributions

T.P.-B. designed and conducted experiments, performed data acquisition, analysis and interpretation, and wrote the manuscript. J.P. carried out experiments, performed data acquisition, analysis and interpretation. P.M. carried out experiments and contributed to data interpretation and drafting of the manuscript. P.L.F. and J.C.T. carried out experiments, performed data acquisition, analysis and interpretation, and participated in the manuscript writing. J.-M.G., A.G. and T.D. conducted experiments and contributed with the drafting of the manuscript. F.B., C.K., S.T. and M.H. conducted experiments and performed analysis of data. J.B.-M. contributed to data analysis and interpretation and edition of the manuscript. G.D. participated in the manuscript writing. E.O. designed experiments, performed interpretation of data, helped with manuscript drafting, and supervised the study. N.C. designed and conducted experiments, performed analysis and interpretation of data, wrote the manuscript, and directed the project.

## Additional information

**Competing interests:** The authors declare no competing financial interests.

