## [Peer Review File · Nature Communications]

Reviewers' comments:

Reviewer #1 (Remarks to the Author):

Cenac and coworkers predict the structures of a GABA and BABA linked lipopeptide from the colibactin pathway based on mass spec and chemical synthesis. One of these metabolites had moderate activity in the inhibition of calcium flux induced by nociceptor activation in sensory neurons. The authors propose that the acyl-appendage promotes passage through the epithelial barrier based on cell assays.

I have some major concerns regarding controls and interpretation of the data, which I outline below. Also, the chemical data is not of the typical standard for the characterization of new metabolites, which leads to some minor skepticism, especially in some metabolites reported in the SI. The moderate activity observed is interesting, although the metabolite's production appears to be about three orders of magnitude lower than what is needed for the observed response.

The paper was generally well written, but I'm not convinced from the data that this metabolite could participate as a functional analgesic associated with pain management in IBS. Either more convincing data should be presented to support publication in Nature Comm, or the authors should tone down their message and submit a more conservative draft to a more specialized journal.

Specific comments

1. Title, Abstract, Classification of metabolites, and "amino-lipid"

For N-myristoyl-D-Asn, "lipoamino acid" or "N-acyl-amide" are the common terms used for classification of these types of molecules.

For N-myristoyl-(likely-D)-Asn-GABA, "lipopeptide" would be a better term from a chemical standpoint. The molecule is an acylated-dipeptide, or more generally, a "lipopeptide," rather than an "amino-lipid."

Also, a μM -level activity does not constitute a "potent" analgesic relative to known potent analgesics.

2. P4. "core machinery consists of" and "machinery also employs"...

Technically, ClbE, ClbG, and ClbQ are types of PKS enzymes as well.

And, the machinery also employs regulatory and resistance proteins.

I'd recommend expanding this section a little or remove the specifics that then lead to an incorrect statement.

3. p5. "Here, we describe ...the isolation and structure elucidation of a new metabolite"

Isolation was not described in the manuscript. Rather, the "planar structure" of the metabolite was characterized by comparisons to a synthetic standard. Please be more specific and do not overstate the chemistry associated with the study. For complete structure elucidation, NMR or X-ray and configurational assignment data for the natural product are typically presented. Stereochemistry was not resolved for their metabolites.

4. p5. "this is the first study characterizing a non-genotoxic bioactive metabolite."

This statement is not completely accurate. Vizcaino et al. reported in JACS (2014) relatively moderate (10 μ M) activity of N-myristoyl-D-Asn (metabolite in Fig 1) against the 5-hydroxytryptamine-7 and Dopamine 5 receptors in vitro. Because inhibition of 5-HT7 is known to alleviate colitis symptoms in DSS-colitis mouse models, the authors speculated that N-myristoyl-D-Asn may be participating in the probiotic response. The authors here also report the concentration of N-myristoyl-D-Asn at 60,000 pg/mg (Fig S6) relative to the 400 pg/mg observed for "C12-Asn-GABA" (Fig S5). In the main text, concentration of C12-Asn-GABA is 50-150 pg/mg, Fig 3.

5. "potent analgesic"

Authors report micromolar level activity. Morphine, for example, binds to receptors in the single digit nanomolar-level range (Kd).

Please delete "potent"

6. p5. "profiles were similar...confirming the chemical composition"

I'd recommend changing "confirming" to "supporting," as multiple solutions are often possible for a molecular formula calculation.

7. p9. "Initial barrier for any drug...is the ...cell wall"

Authors should indicate that this is after penetrance of the mucosal barrier.

8. p9. "GABA was not detected in the basal chamber...did not cross the intestinal epithelial monolayer"

The comparison of GABA to C12-Asn-GABA is a key component of the manuscript. The detection limits of GABA in Fig. 5B look to be very low. Could GABA be metabolized in the basal area by the 24 h time point, providing an alternative explanation for the observed data? A time course could be helpful to resolve this issue.

9. Again, GABA to C12-Asn-GABA comparison is a key component of the manuscript. P10. "C12-Asn-GABA" to cross the epithelial barrier, in vivo"

It would be helpful to compare free GABA. Does it really not cross in vivo? See also comment 8.

10. Minor fix. P7. "trans-acyl-transferase" should be "trans-acyl-transferase PKS"

The two terms represent completely different enzyme functions.

11. p11. "C12-Asn-GABA requires three enzymes"

Authors show an absolute requirement for ClbN and maybe a requirement for ClbB for the C12-Asn-BABA. ClbA reduces production but is not required. Please be more specific or clarify.

Proposed structure of C12-Asn-BABA doesn't make sense from a PKS-NRPS perspective. Adding comparative tandem-MS and co-injection experiments would be more supportive. NMR would be better yet. The similar retention time shown is suggestive but not absolute.

12. p12. "This study is the first step...bioactive amino lipids ...by the gut microbiota"

Authors should not overstate their findings. Brady et al. reported in PNAS (2015) how "lipoamino acids"/"N-acyl-amides" produced in commensal organisms regulate inflammation. They called this

metabolite "commendamide."

Figures

13. Figure 1 could just as easily be a supporting Figure, as this metabolite has been described many times at this point. Inactivation of pantetheinyltransferase ClbA would as expected minimize ClbN activation needed to make this known metabolite. Also, stereochemistry is known for this metabolite, which could be shown.

14. Figure 2 should provide stereochemistry, and stereochemistry should be defined in the synthetic methods. This reviewer could not assess the full structural characterization, as the methods did not include stereochem.

For example, are there two C12-Asn-AABA peaks because the authors synthesized a pair of diastereomers? This question is also relevant for C12-Asn-BABA.

15. Figure 3.

The authors report C12-Asn-GABA at ~100 pg/mL.

When converted to uM, this equals 2.5×10^{-4} uM.

When converted to nM, this equals 0.25 nM.

Unfortunately, I found this figure to be a strong piece of data arguing against the authors' hypotheses.

In subsequent Fig 4, they show that the C12-Asn-GABA is a moderate inhibitor (uM range). Consequently, the production is three orders of magnitude lower than what is needed for an inhibitory response.

16. Figure 5. Intensity (AU)

Define units. This was confusing. Absorbance Units? milli-Absorbance Units (mAU)? Arbitrary Units (then how was this calculated)? Something else?

Confidence in GABA detection in the apical department could not be assessed by this reviewer due to a lack of understanding of the units. Detection limits should be good for subsequent comparison in the basal compartment.

Supporting Information

Fig S1. ClbP cleavage reaction is incorrect.

Fig S2-S3. Proposed leucyl-analog shown could easily be a different structure based on the data presented.

Fig. S4. Show stereochem.

Reviewer #2 (Remarks to the Author):

Review of Pérez-Berezo et al., "Identification of an analgesic amino-lipid produced by the probiotic *Escherichia coli* strain Nissle 1917"

This is a novel and interesting study that has high potential significance within the field. It uses a wide array of interdisciplinary approaches to characterise a novel analgesic ligand produced by a probiotic that is used to treat several intestinal disorders.

The methodologies used are state of the art and appropriate. The interpretation of the findings is sound. The statistics appear appropriate as does the number of replicates used in the studies.

I do have a few questions/clarifications:

1. How do the concentrations of C12-Asn-GABA used in the calcium flux studies correlate with those detected in vivo?
2. Do the authors have evidence for expression of GABA B receptors on nociceptors. The authors show calcium influx is reduced by GABA B receptor activation. Is there evidence for expression of these channels with GABA B? Is there appropriate evidence in the literature?
3. Can the authors speculate as to the key mechanism by which the addition of C12-Asn means that C12-Asn-GABA but not GABA can cross epithelial cells? Is it pure a diffusion process or is a pump required?
4. Is the anti-nociceptive action purely mediated by C12-Asn-GABA, or is it subsequently metabolized?
5. Representative examples of the visceral hypersensitivity studies should be presented.
6. Why was duodenal motility studied when visceral hypersensitivity was assessed in response to intra-colonic administration?
7. The numbering of the Figure 6 legend does not appear to correlate with the data shown.

Reviewer #3 (Remarks to the Author):

This is a very interesting manuscript describing one of the products of the colibactin (PKS) cluster from *E. coli* Nissle 1917. The key finding is in figure 2 where high resolution mass spectrometry was used to identify a metabolite from the pathway. The metabolite is a cleavage product of the colibactin biosynthesis pathway. Mass spec analysis showed that the metabolite is C12-asparagine-aminobutyric acid. Since mass spec alone is insufficient to identify the isomer of butyric acid linked to the fatty acid in figure 2D the authors synthesized several authentic references and showed the presence of only C12-Asn-GABA and C12-Asn-BABA. I believe the synthesis of the authentic samples coupled with the mass spec data provides convincing evidence of the structure proposed. Overall I believe the data supports the conclusion of the paper. I did notice two minor issues in the manuscript.

1. Page 7, line 161: (fig 3B and 3C) should be (fig 3A and 3B).
2. Supp figure 1: In figure legend, the authors mentioned C14-Asn as the cleavage product but in the figure, the C12-Asn is drawn.

Reviewers' comments:

Reviewer #1 (Remarks to the Author):

Cenac and coworkers predict the structures of a GABA and BABA linked lipopeptide from the colibactin pathway based on mass spec and chemical synthesis. One of these metabolites had moderate activity in the inhibition of calcium flux induced by nociceptor activation in sensory neurons. The authors propose that the acyl-appendage promotes passage through the epithelial barrier based on cell assays.

I have some major concerns regarding controls and interpretation of the data, which I outline below. Also, the chemical data is not of the typical standard for the characterization of new metabolites, which leads to some minor skepticism, especially in some metabolites reported in the SI. The moderate activity observed is interesting, although the metabolite's production appears to be about three orders of magnitude lower than what is needed for the observed response.

The paper was generally well written, but I'm not convinced from the data that this metabolite could participate as a functional analgesic associated with pain management in IBS. Either more convincing data should be presented to support publication in Nature Comm, or the authors should tone down their message and submit a more conservative draft to a more specialized journal.

Specific comments

1. Title, Abstract, Classification of metabolites, and "amino-lipid"

For N-myristoyl-D-Asn, "lipoamino acid" or "N-acyl-amide" are the common terms used for classification of these types of molecules.

For N-myristoyl-(likely-D)-Asn-GABA, "lipopeptide" would be a better term from a chemical standpoint. The molecule is an acylated-dipeptide, or more generally, a "lipopeptide," rather than an "amino-lipid."

In agreement with the reviewer's comment, the appropriate term has been used along the manuscript

Also, a μM -level activity does not constitute a "potent" analgesic relative to known potent analgesics.

For somatic pain and even more for visceral pain, the dose used to activate receptors on transfected cells are lower than the dose needed *in vivo* to obtain an analgesic effect. In our study, we demonstrated an *in vivo* analgesic effect of the C12AsnGABAOH in the micromolar range as it may be observed for morphine (1-3). Furthermore, in primary culture of sensory neurons *in vitro*, morphine, DPDPE (delta-opioid receptor agonist), SNC80 (opioid receptor agonist) or DAMGO (μ -opioid receptor agonist) are reported to be active in a micromolar range as the C12AsnGABAOH used in our study (4-7). We also performed additional experiments to test the analgesic properties of GABA alone in the same condition. We showed that pretreatment of primary culture of sensory neurons with GABA (fig. S8) and C12AsnGABAOH (fig. 4) decreases capsaicin-induced calcium flux in a micromolar level, as shown for the Baclofen a GABAB receptor agonist (8). However, as we did not assess the *in vivo* analgesic effect of our compound at the somatic level, the term visceral has been added after potent (potent visceral analgesics) in the manuscript.

1. M. B. Burton and G. F. Gebhart. Effects of Kappa-Opioid Receptor Agonists on Responses to Colorectal Distension in Rats with and without Acute Colonic Inflammation. *JPET* 285:707–715, 1998
2. Veysel Baskin, S Sirri Bilge, Ayhan Bozkurt, Bahar Akyuz, Arzu Erdal Agri, Hasan Guzel, Fatih Ilkaya. Effect of nonsteroidal anti-inflammatory drugs on colorectal distension-induced visceral pain. *Indian J Pharmacol.* 2016 Mar-Apr;48(2):150-4.
3. Jyoti N. Sengupta, Xin Su, and Gerald F. Gebhart. κ , But Not μ or δ , Opioids Attenuate Responses to Distention of Afferent Fibers Innervating the Rat Colon. *Gastroenterology* 1996;111:968 – 980
4. Thiago M. Cunha, Danilo Roman-Campos, Celina M. Lotufo, Hugo L. Duarte, Guilherme R. Souza, Waldiceu A. Verri, Jr., Mani I. Funez, Quintino M. Dias, Ieda R. Schivo, Andressa C. Domingues, Daniela Sachs, Silvana Chiavegatto, Mauro M. Teixeira, John S. Hothersall, Jader S. Cruz, Fernando Q. Cunha, and Sergio H. Ferreira. Morphine peripheral analgesia depends on activation of the PI3K γ /AKT/nNOS/NO/K_{ATP} signaling pathway. *Proc Natl Acad Sci U S A.* 2010 Mar 2;107(9):4442-7.
5. Cai Q, Qiu CY, Qiu F, Liu TT, Qu ZW, Liu YM, Hu WP. Morphine inhibits acid-sensing ion channel currents in rat dorsal root ganglion neurons. *Brain Res.* 2014 Mar 20;1554:12-20.
6. Wu ZZ, Chen SR, Pan HL. Distinct inhibition of voltage-activated Ca²⁺ channels by delta-opioid agonists in dorsal root ganglion neurons devoid of functional T-type Ca²⁺ currents. *Neuroscience.* 2008 Jun 2;153(4):1256-67.
7. Hai-Bo Wang, BoZhao, Yan-Qing Zhong, Kai-Cheng Li, Zi-YanLi, QiongWang, Yin-Jing Lu, Zhen-Ning Zhang, Shao-Qiu He, Han-Cheng Zheng, Sheng-XiWu, Tomas G. M. Hökfelt, LanBao, and Xu Zhang. Coexpression of δ - and μ -opioid receptors in nociceptive sensory neurons. *Proc Natl Acad Sci U S A.* 2010 Jul 20; 107(29):13117-22.
8. Hanack C, Moroni M, Lima WC, Wende H, Kirchner M, Adelfinger L, Schrenk-Siemens K, Tappe-Theodor A, Wetzell C, Kuich PH, Gassmann M, Roggenkamp D, Bettler B, Lewin GR, Selbach M, Siemens J. GABA blocks pathological but not acute TRPV1 pain signals. *Cell.* 2015 Feb 12;160(4):759-70.

2. P4. “core machinery consists of” and “machinery also employs”... Technically, ClbE, ClbG, and ClbQ are types of PKS enzymes as well. And, the machinery also employs regulatory and resistance proteins. I’d recommend expanding this section a little or remove the specifics that then lead to an incorrect statement.

We have corrected this section adding the fact that “The machinery also employs additional maturation proteins, smaller enzymes with modules of PKS enzyme, an efflux pump, a resistance protein and a putative regulatory protein.”

3. p5. “Here, we describe ...the isolation and structure elucidation of a new metabolite” Isolation was not described in the manuscript. Rather, the “planar structure” of the metabolite was characterized by comparisons to a synthetic standard. Please be more specific and do not overstate the chemistry associated with the study. For complete

structure elucidation, NMR or X-ray and configurational assignment data for the natural product are typically presented. Stereochemistry was not resolved for their metabolites.

We do agree with the reviewer, “isolation” has been suppressed from our manuscript. Following the Editorial & publishing policies of nature communications for the characterization of chemical and biomolecular materials, ¹H-NMR and ¹³C-NMR has been performed on the synthesized standards and the compound characterization data, formatted according to nature communications policy, have been added in the supplementary material in the section: “Synthesis of C12-Asn-alpha aminobutyric acid, C12-Asn-beta aminobutyric acid and C12-Asn-gamma aminobutyric acid”. Moreover, images of spectral data of the C12AsnGABAOH, C12AsnBABAOH and C12AsnAABAOH have been included as supplementary figure 10, 11 and 12. In Figure 2D, we added the stereochemistry of the synthesized compounds described in the main manuscript line 173. Also the stereochemistry has been included in the figure S3 which described the syntheses and in the text.

To make the analysis either by NMR or/and by X-ray of the natural product, the sample amount required is more than several tens of µg and several mg, respectively. In our culture condition, we have a production of 200 pg in bacterial pellet of culture (10 mL); thus, in order to reach the amount necessary for NMR, we will have to perform a culture of 100 L and to extract the bacterial pellets. In our preliminary experiments, we determine that in our conditions we can not extract bacterial pellet coming from more than 50 mL culture, otherwise we have a significant decrease of extraction efficacy. To reach the amount needed, we will have to perform at least 2000 extractions. Moreover, purification of our compound by liquid chromatography will need about 10 000 injections and as C12AsnGABAOH is not the only compound on the chromatogram (several other molecules are close to its retention time), there is no guarantee to get the pure molecule which is needed to perform an efficient NMR characterization. Obtaining enough natural compounds under our conditions will be highly time-consuming with no guarantee of success. Thus, we decide to try to increase the production of C12AsnGABAOH in our culture.

Although colibactin, the final product of the biosynthetic machinery encoded by the *pks* island has not been purified and characterized so far, we tried to increase yield of C12-Asn-GABA by using a strain deleted for *clbP* based on the publication by M. I. Vizcaino and J. M. Crawford (ref 19 of our manuscript). As previously published, we confirmed a 50 times increase of C14AsnVal by LC-MS/MS in mutated bacteria (65 pg/mg of protein in wild type EcN and 3259 pg/mg of protein in Δ*clbP* EcN) but in contrast to the cleavage product of precolibactin, this mutation has not effect on C12AsnGABA synthesis (fig. 3). In order to increase the production of C12AsnGABAOH, we tried several culture conditions. In a first set of experiments, we cultured our bacteria in presence of GABA in the minimal medium A, the concentration of C12AsnGABAOH was only increased by 1.2 times; thus this condition did not have a significant effect on the compound production. Then, we tried several media but we failed to increase the production of C12AsnGABAOH. In LB medium, we quantified C14AsnGABAOH instead of C12AsnGABAOH. To determine the physiological production of these metabolites, we quantified C14AsnGABAOH and C12AsnGABAOH concentration in different mouse organs by LC-MS/MS. Although C14AsnGABAOH is more produce in LB medium than the C12AsnGABAOH, we were not able to quantify the C14AsnGABAOH in colon, caecum, blood or brain of a mouse in contrast to the C12AsnGABAOH (fig. 6 and fig. below).

In conclusion, we did not find a culture condition to increase the production of C12AsnGABAOH in order to be able to determine the structure by RMN. Nevertheless, we performed the structural determination (retention time and fragmentation patterns) of the

compounds by LC / ESIMS/MS and obtained exactly similar data compared to the standard compounds (whose structures were confirmed by NMR investigation). In metabolomics and in lipidomics, the structure of a metabolite considered as *de novo* is identical (except in terms of chirality) to that of the standard compound (commercial or synthetic with the structure confirmed by NMR) when they possess several identical analytical characteristics (1-3). Among these, the most significant are:

- (i) the same LC / MS retention time for separation using the same chromatograph (in terms of column, solvents, and buffer) coupled to the mass spectrometer,
- (ii) an elementary composition for the charged molecular species determined with some accuracy better than 5 ppm, and
- (iii) after activation of this species selected in MS / MS, a same i.e. fingerprint (i.e., in terms of m/z ratios for the produced ions with a same elementary composition and product ion abundance) to that characterizing the standard.

In the previous version of our article we achieved two criteria. In fact, the retention times between the molecules identified in our samples and the standard was similar: 5.34 min for the C12AsnGABAOH and 5.68 min for the C12AsnBABAOH (fig. 2D). Moreover, the experimental ratio mass/charge obtained at 398.2664 was at 0.6 ppm from the theoretical ratio mass/charge at 398.2660 (fig. 2). Thus, in order to complete the identification criteria of the metabolomics rules, we determined the fragmentation spectra on the high resolution mass spectrometer QExactive+ (Thermo) of the identified molecules in our samples and of the 4 synthesized standard: C12AsnGABAOH, C12AsnBABAOH and C12AsnAABAOH (2 isomers) at two normalized collision energies (NCE) 20 and 35%. Fragmentation spectra are presented on fig S9. We then characterize all the fragments generated during the fragmentation experiment by the determination of the elementary composition (supplementary table 1). These results allowed us to elucidate the mechanism of specific fragments of each aminobutyric isomer (Supplementary scheme 2, 3 and 4 and supplementary results) and to confirm the metabolites C12AsnBABAOH and C12AsnGABAOH. The m/z ratios for the produced ions of the samples are similar to the produced ions of the standard. In conclusion, we reach the metabolomics and lipidomics criteria for molecular identification. These novel data have been added in the main manuscript in the paragraph: "Identification of lipoamino acid and lipopeptides produced by EcN" and in the supplementary scheme 2, 3 and 4, supplementary results, supplementary table 1 and supplementary figure S9.

1- Peironcely JE, Rojas-Chertó M, Tas A, Vreeken R, Reijmers T, Coulier L, Hankemeier T. Automated pipeline for de novo metabolite identification using mass-spectrometry-based metabolomics. *Anal. Chem.* 2013, 85, 3576 -3583

2- Rathahao-Paris, E.; Alves, S.; Junot, C.; Tabet, J.-C. High resolution mass spectrometry for structural identification of metabolites in metabolomics. *Metabolomics* 2016, 12, 1-15.

3- Sumner LW, Amberg A, Barrett D, Beale MH, Beger R, Daykin CA, Fan TW, Fiehn O, Goodacre R, Griffin JL, Hankemeier T, Hardy N, Harnly J, Higashi R, Kopka J, Lane AN, Lindon JC, Marriott P, Nicholls AW, Reily MD, Thaden JJ, Viant MR. Proposed minimum reporting standards for chemical analysis Chemical Analysis Working Group (CAWG) Metabolomics Standards Initiative (MSI). *Metabolomics*. 2007 Sep;3(3):211-221.

Quantification of C12AsnGABA in mouse organs by LC-MS/MS as described in our manuscript.

4. p5. “this is the first study characterizing a non-genotoxic bioactive metabolite.”

This statement is not completely accurate. Vizcaino et al. reported in JACS (2014) relatively moderate (10 μ M) activity of N-myristoyl-D-Asn (metabolite in Fig 1) against the 5-hydroxytryptamine-7 and Dopamine 5 receptors *in vitro*. Because inhibition of 5-HT7 is known to alleviate colitis symptoms in DSS-colitis mouse models, the authors speculated that N-myristoyl-D-Asn may be participating in the probiotic response. The authors here also report the concentration of N-myristoyl-D-Asn at 60,000 pg/mg (Fig S6) relative to the 400 pg/mg observed for “C12-Asn-GABA” (Fig S5). In the main text, concentration of C12-Asn-GABA is 50-150 pg/mg, Fig 3.

In agreement with the reviewer comment, the sentence has been modified: “...this is the first study characterizing a non-genotoxic bioactive metabolite *in vivo*.”

5. “potent analgesic”

Authors report micromolar level activity. Morphine, for example, binds to receptors in the single digit nanomolar-level range (Kd).

Please delete “potent”

As developed in the first comment, several studies have shown that analgesics -including morphine-decrease neuronal activation *in vitro*, or inhibit *in vivo* somatic or visceral pain in a micromolar range. The term visceral has been added after potent (potent visceral analgesics) in the manuscript.

6. p5. “profiles were similar...confirming the chemical composition”

I’d recommend changing “confirming” to “supporting,” as multiple solutions are often possible for a molecular formula calculation.

Following the referee’s recommendation, “confirming” has been replaced by “supporting”.

7. p9. “Initial barrier for any drug...is the ...cell wall”

Authors should indicate that this is after penetrance of the mucosal barrier.

In agreement with the reviewer, the sentence has been modified as follows: “The initial barrier for any drug absorption is the intestinal epithelial cell wall after penetrance of the mucus layer.”

8. p9. “GABA was not detected in the basal chamber...did not cross the intestinal epithelial monolayer”

The comparison of GABA to C12-Asn-GABA is a key component of the manuscript. The detection limits of GABA in Fig. 5B look to be very low. Could GABA be metabolized in

the basal area by the 24 h time point, providing an alternative explanation for the observed data? A time course could be helpful to resolve this issue.

In accordance with the reviewer's comment, a kinetic was performed using a new method on an Orbitrap Q Exactive plus mass spectrometer (supplementary method paragraph: "GABA analysis by LC-HRMS". For an accurate quantification of the GABA, an internal standard corresponding to U-13C-GABA and calibration curve were used. In addition, the limit of quantification and the limit of detection were determined in culture media; the LOQ was 1.73 μ M and the LOD 0.43 μ M. The quantity of GABA in the basolateral compartment was under the LOD at the 3 time points tested. Our results demonstrated that the GABA is not able to cross the epithelial barrier from 3 to 16 hours after its treatment at the apical side (Fig. 5B).

9. Again, GABA to C12-Asn-GABA comparison is a key component of the manuscript. P10. "C12-Asn-GABA" to cross the epithelial barrier, in vivo" It would be helpful to compare free GABA. Does it really not cross in vivo? See also comment 8.

Using the method describe above, GABA has been quantified in blood and colon tissue, 30 min after its intracolonic administration. In contrast to C12AsnGABA OH , the GABA alone is not able to cross the epithelial barrier *in vivo* (Fig. 6C).

10. Minor fix. P7. "trans-acyl-transferase" should be "trans-acyl-transferase PKS" The two terms represent completely different enzyme functions.

The modification has been made according to the reviewer's comment.

11. p11. "C12-Asn-GABA requires three enzymes"

Authors show an absolute requirement for ClbN and maybe a requirement for ClbB for the C12-Asn-BABA. ClbA reduces production but is not required. Please be more specific or clarify.

In agreement with the reviewer comment, the sentence has been modified as follow: "Three enzymes implicated in the synthesis of this C12AsnGABA OH have been identified. Surprisingly, these enzymes were encoded by a genomic island, named *pks*, which carries the cluster of genes that enables the synthesis of hybrid peptide polyketides and especially the genotoxin colibactin, a *bona fide* virulence factor and a putative carcinogenic agent."

Proposed structure of C12-Asn-BABA doesn't make sense from a PKS-NRPS perspective. Adding comparative tandem-MS and co-injection experiments would be more supportive. NMR would be better yet. The similar retention time shown is suggestive but not absolute.

As developed in the response of the 3rd comment, in the previous version of our article we achieved two criteria for the structural elucidation by mass spectrometry, the same retention time and the experimental ratio mass/charge obtained at 0.6 ppm from the theoretical ratio mass/charge (fig. 2). In addition, we determined the fragmentation spectra on high resolution mass spectrometer of the identified molecules in our samples and of the synthesized C12AsnBABA OH (fig S9) and characterized all the fragments generated (supplementary table 1). These results allowed us to confirm the metabolites C12AsnBABA OH in our samples.

**12. p12. “This study is the first step...bioactive amino lipids ...by the gut microbiota”
Authors should not overstate their findings. Brady et al. reported in PNAS (2015) how
“lipoamino acids”/”N-acyl-amides” produced in commensal organisms regulate
inflammation. They called this metabolite “commendamide.”**

In agreement with the reviewer’s comment, this sentence has been removed from the manuscript.

Figures

13. Figure 1 could just as easily be a supporting Figure, as this metabolite has been described many times at this point. Inactivation of pantetheinyltransferase ClbA would as expected minimize ClbN activation needed to make this known metabolite. Also, stereochemistry is known for this metabolite, which could be shown.

We agree with the referee that this metabolite has been widely described. However, even if the metabolite in Figure 1 has already been published, this figure is the starting point of the C12AsnGABA OH identification. In fact, on this figure the total ion chromatogram of EcNwt and EcN Δ clbA is represented and show the differences in lipid expression in these two strains. Moreover, the product ions used for the characterization of the C12AsnGABA OH are also represented on this figure. Thus, for the reader, we think that it is important to include this figure in the main manuscript to follow the experimental approach leading to the characterization of the compound of interest.

14. Figure 2 should provide stereochemistry, and stereochemistry should be defined in the synthetic methods. This reviewer could not assess the full structural characterization, as the methods did not include stereochem.

For example, are there two C12-Asn-AABA peaks because the authors synthesized a pair of diastereomers? This question is also relevant for C12-Asn-BABA.

In agreement with the reviewer comment, the stereochemical structures of the synthesized compounds have been added in figure 2D and mentioned in the legend of the figure 2 and in the paragraph “Identification of lipoamino acid and lipopeptides produced by EcN”. In addition, the stereochemistry details have been included in the figure S3 which described the syntheses. In paragraph “Synthesis of C12-Asn-alpha aminobutyric acid, C12-Asn-beta aminobutyric acid and C12-Asn-gamma aminobutyric acid” of the supplementary materials, we detailed the chiral pools used to make those compounds and added the 1H -NMR and ^{13}C -NMR compound characterization. Images of spectral data of the C12AsnGABA OH , C12AsnBABA OH and C12AsnAABA OH have been included as supplementary figure 10, 11 and 12.

For the C12AsnAABA OH , we observed epimerization during the synthesis as described in the different section cited above. Epimerization was observed only during the synthesis of C12AsnAABA OH and it occurred at the AABA moiety during the coupling of the AABA derivative with Na-Fmoc-Ng-trityl-L-asparagine and not during the last coupling with the fatty acid. Therefore, BABA being not an alfa amino acid like AABA, epimerization during C12AsnBABA synthesis did not occur, and as such only one isomer is observed.

15. Figure 3.

The authors report C12-Asn-GABA at ~100 pg/mL.

When converted to uM, this equals 2.5×10^{-4} uM.

When converted to nM, this equals 0.25 nM.

Unfortunately, I found this figure to be a strong piece of data arguing against the authors' hypotheses.

In subsequent Fig 4, they show that the C12-Asn-GABA is a moderate inhibitor (uM range). Consequently, the production is three orders of magnitude lower than what is needed for an inhibitory response.

As mentioned in the response to the first comment, in primary culture of sensory neurons, classical analgesic drugs are used in a micromolar range concentration needed for the inhibition of nociceptor. Based on the literature, the concentration of C12AsnGABA OH used in this study is in the range of the concentration of classical potent inhibitors such as morphine, GABA, DPDPE or baclofen used for calcium flux experiments in primary culture of sensory neurons.

The concentration of 100 pg/mL of C12AsnGABA OH is the concentration in bacterial culture performed in 10 mL of minimal medium A, which is probably completely different from the concentration produced *in vivo* by the bacteria present in the colon.

For a work in progress in our laboratory we added several lipopeptides to our method of LC-MS/MS (C14Asn, C16Asn, C14AsnLeu...) to quantify these metabolites in mice. Even if the C14Asn was the most concentrated lipoamino acid in cultured ECN bacteria, in C57Bl6 mice we were unable to quantify C14Asn in the colonic tissue, in the brain, in the liver or in the blood, in contrast to the C12AsnGABA (fig 6A and 6B white bars for the blood and the colon). To assess the effect of the culture conditions on lipopeptide production by EcN, we quantified those compounds in EcN cultured in LB medium. Under this condition, we were not able to quantify C12AsnGABA OH . In contrast, we quantified between 2 to 4 ng of C14AsnGABA OH . This compound, which differs from the molecule described in our study only by two carbons, was not quantifiable in mice.

All these results demonstrate the difficulty to extrapolate a dose or even a compound quantified in cultured bacteria to the *in vivo* condition. More research from our lab and others is needed to characterize the quantity and the nature of lipopeptide produced *in vivo* by *E. coli*.

16. Figure 5. Intensity (AU)

Define units. This was confusing. Absorbance Units? milli-Absorbance Units (mAU)?

Arbitrary Units (then how was this calculated)? Something else?

Confidence in GABA detection in the apical department could not be assessed by this reviewer due to a lack of understanding of the units. Detection limits should be good for subsequent comparison in the basal compartment.

The figure representing the quantification has been redrawn in accordance with the method newly developed.

Supporting Information

Fig S1. ClbP cleavage reaction is incorrect.

The cleavage reaction has been corrected.

Fig S2-S3. Proposed leucyl-analog shown could easily be a different structure based on

the data presented.

The data presented have been improved in order to support the proposed leucyl-analog (supplementary figure 2 and in the main manuscript). In particular the HCD spectra of Leu or Ileu linked to CnAsn moiety can be distinguished by the reproducible variation in product ion abundances of several small abundant ions and on the stability of the precursor ion (will be published elsewhere).

Fig. S4. Show stereochem.

The stereochemistry has been included in the figure which is supplementary figure 3 in the revised version and in figure 2.

Reviewer #2 (Remarks to the Author):

Review of Pérez-Berezo et al., “Identification of an analgesic amino-lipid produced by the probiotic *Escherichia coli* strain Nissle 1917”

This is a novel and interesting study that has high potential significance within the field. It uses a wide array of interdisciplinary approaches to characterize a novel analgesic ligand produced by a probiotic that is used to treat several intestinal disorders.

The methodologies used are state of the art and appropriate. The interpretation of the findings is sound. The statistic appears appropriate as does the number of replicates used in the studies.

I do have a few questions/clarifications:

1. How do the concentrations of C12-Asn-GABA used in the calcium flux studies correlate with those detected in *in vivo*?

In vivo, 30 minutes after the intracolonic administration of C12AsnGABAOH we quantified 7 ng of this compound in a piece of tissue corresponding to 1 cm of colon. Concerning our calcium flux study *in vitro*, as the molecular weight of our compound is 400 g/mol and the volume of our working medium is 150 μ L, 1 μ M would correspond to 80 ng. Thus, the quantity found in a small piece of colon 30 min after intracolonic administration of C12AsnGABAOH may correspond to the 0.1 μ M concentration *in vitro*. The concentrations are not perfectly correlated but are both in the micromolar range.

2. Do the authors have evidence for expression of GABA B receptors on nociceptors. The authors show calcium influx is reduced by GABA B receptor activation. Is there evidence for expression of these channels with GABA B? Is there appropriate evidence in the literature?

Numerous studies demonstrate that the GABAB receptors are robustly expressed in peripheral nociceptive neurons highlighting an unrecognized role for GABAB receptors in the pain pathway (1, 2). More accurately in the spinal cord, GABAB receptors were expressed in the superficial layers of the dorsal horn, as well as in motor neurons in the deeper layers of the ventral horn (1). Moreover, the GABAB receptor subunits GABAB1 and GABAB2 mRNA and the corresponding subunit proteins are present in dorsal root ganglia (2). Hanack and coworkers using immunocytochemistry and bioluminescence resonance energy transfer studies showed that GABAB receptor in particular the subunit GABAB1 and TRPV1 channel co-localize at peripheral nerve endings and sensory neurons of mice, suggesting that GABA might decrease capsaicin-induced calcium flux (3). To reinforce this purpose, in this

study, the pretreatment of sensory neurons with baclofen -a GABAB receptor agonist- strongly blocked TRPV1 channel sensitization and hyperactivity induced by inflammatory mediators, which suggests the existence of crosstalk between them (3).

1. Charles, K.J., Evans, M.L., Robbins, M.J., Calver, A.R., Leslie, R.A., and Pangalos, M.N. (2001). Comparative immunohistochemical localisation of GABA(B1a), GABA(B1b) and GABA(B2) subunits in rat brain, spinal cord and dorsal root ganglion. *Neuroscience* 106, 447–467.

2. Towers, S., Princivalle, A., Billinton, A., Edmunds, M., Bettler, B., Urban, L., Castro-Lopes, J., and Bowery, N.G. (2000). GABAB receptor protein and mRNA distribution in rat spinal cord and dorsal root ganglia. *Eur. J. Neurosci.* 12, 3201–3210.

3. Hanack C, Moroni M, Lima WC, Wende H, Kirchner M, Adelfinger L, Schrenk-Siemens K, Tappe-Theodor A, Wetzel C, Kuich PH, Gassmann M, Roggenkamp D, Bettler B, Lewin GR, Selbach M, Siemens J. GABA blocks pathological but not acute TRPV1 pain signals. *Cell.* 2015;160:759–770.

3. Can the authors speculate as to the key mechanism by which the addition of C12-Asn means that C12-Asn-GABA but not GABA can cross epithelial cells? Is it pure a diffusion process or is a pump required?

We performed additional experiments to reinforce this point. In fact, we believe that the key mechanism is that the addition of C12Asn allows the passage of the GABA through the epithelium. We developed a new method for the quantification of GABA using U-13C-GABA as an internal standard to determine the limit of quantification and the limit of detection (supplementary method). *In vitro*, we performed kinetics and quantified the GABA 3, 6 and 16 hours after its treatment at the apical side of Caco2 cells cultured in transwell. For these time points, GABA was not quantifiable at the basolateral side (fig. 5B). Moreover, using this methodology, we assessed the passage of GABA *in vivo* by an intracolonic administration of GABA and its quantification 30 min later in the colon and in the blood. The concentration of GABA was not increased in either the blood or colon after the treatment (fig. 6C). These two experiments demonstrate that GABA alone is not able to cross the intestinal epithelium in physiological conditions. In contrast, *in vitro* and *in vivo*, C12AsnGABAOH was able to cross the epithelium (fig. 5A and fig. 6A and 6B).

It is really difficult to determine if it is a pure diffusion process or if a pump is required. As GABA was not able to cross the epithelium, we hypothesize that if it was an active passage, it would be linked to the fatty acid. The fatty acid translocase (FAT)/CD36 (now designated as SR-B2) is the main membrane protein involved in fatty acid uptake into intestinal enterocytes. Nevertheless, as we did not find expression of CD36 on Caco2 cells by Rt-qPCR, we think that the passage is probably pure diffusion. Several experiments, in which authors are absolutely not specialist, will be needed to determine the mechanism of the passage of C12AsnGABAOH across the epithelium.

4. Is the anti-nociceptive action purely mediated by C12-Asn-GABA, or is it subsequently metabolized?

In order to answer to the reviewer comment, we performed BRET experiments with GABAB receptor transiently expressed in HEK293 cells in collaboration with Dr. Julie Kniazeff and Dr. Cyril Goudet of the Institute for Functional Genomics, Montpellier, France. As shown in the figure below, GABA induced a decrease in BRET ratio induced by the dissociation of Gao/ γ 2 subunits following receptor activation in GABAB transfected cells

(solid symbols) and not in cells transfected with an empty vector (Mock; empty symbols). In contrast, C12AsnGABA did not induce a decrease in BRET ratio. These results demonstrates that C12AsnGABA do not activate directly the GABAB receptor. Thus, C12AsnGABAOH needs to be metabolized to activate GABAB receptor.

	GABA _B -GABA
Bottom	230.5
Top	285.9
LogIC50	-6.738
HillSlope	-1.038
IC50	1.827e-007
Span	55.38

Following these experiments, we quantified the C12Asn 5, 10, 30 and 60 min after the treatment of primary culture of sensory neurons with 10 μ M of C12AsnGABA. The quantity of C12Asn increased overtime (see figure below). Thus, it seems that the C12AsnGABA is cleaved by protease(s) expressed by the neurons or glial cells to release GABA. To confirm, this result we should need to develop a new methodology to quantify C12AsnGABA, C12Asn and GABA in the same sample.

Nevertheless, as we did not identify the protease(s) and several experiments and method development would be needed to confirm our hypothesis, we did not add these results to the manuscript. Moreover, we cannot exclude also an interaction of our compounds with

transporters of the GABA which are highly regulated. We are currently continuing this collaboration and experiments in order to decipher the mechanism leading to GABAB receptor activation by the C12AsnGABA.

5. Representative examples of the visceral hypersensitivity studies should be presented.

Representative traces of mouse abdominal muscle contractions in response to 60 mm Hg colorectal distension have been added in the figure 6D.

6. Why was duodenal motility studied when visceral hypersensitivity was assessed in response to intra-colonic administration?

The motility of the colon is characterized by spontaneous waves that permit the contraction of intestinal muscles. In the colon, contraction waves are asynchronous (1) which render difficult the exact quantification of mechanical intestinal contractions in terms of amplitude and/or frequency. As opposed to the colon, the duodenum presents regular contraction waves. Using isotonic sensors, studies from our team clearly shown that this *ex vivo* method is particularly adapted to the measurement of regular and pulsatile signals (2). In the present study, the choice of the duodenum to test the effect of C12AsnGABAOH on motility is justified by the fact that the potential effect of this lipid on intestinal contraction could be masked in the colon. The facilitation of recording the extremely regulated contraction of the duodenum has confirmed our choice.

1. M. Kocylowski, K. L. Bowes, and Y. J. Kingma. Electrical and Mechanical Activity in the Ex Vivo Perfused Total Canine Colon. *Gastroenterology*; 77:1021-1026,1979

2. Fournel A, Drougard A, Duparc T, Marlin A, Brierley SM, Castro J, Le-Gonidec S, Masri B, Colom A, Lucas A, Rousset P, Cenac N, Vergnolle N, Valet P, Cani PD, Knauf C. Apelin targets gut contraction to control glucose metabolism via the brain. *Gut*. 2017 Feb;66(2):258-269.

7. The numbering of the Figure 6 legend does not appear to correlate with the data shown.

The numbering of Figure 6 legend has been rewritten.

Reviewer #3 (Remarks to the Author):

This is a very interesting manuscript describing one of the products of the colibactin (PKS) cluster from *E. coli* Nissle 1917. The key finding is in figure 2 where high resolution mass spectrometry was used to identify a metabolite from the pathway. The metabolite is a cleavage product of the colibactin biosynthesis pathway. Mass spec analysis showed that the metabolite is C12-asparagine-aminobutyric acid. Since mass spec alone is insufficient to identify the isomer of butyric acid linked to the fatty acid in figure 2D the authors synthesized several authentic references and showed the presence of only C12-Asn-GABA and C12-Asn-BABA. I believe the synthesis of the authentic samples coupled with the mass spec data provides convincing evidence of the structure proposed. Overall I believe the data supports the conclusion of the paper. I did notice two minor issues in the manuscript.

1. Page 7, line 161: (fig 3B and 3C) should be (fig 3A and 3B).

The correction has been made in the manuscript.

2. Supp figure 1: In figure legend, the authors mentioned C14-Asn as the cleavage product but in the figure, the C12-Asn is drawn.
The cleavage reaction has been corrected.

REVIEWERS' COMMENTS:

Reviewer #1 (Remarks to the Author):

The revision is much, much stronger, and in my opinion, the revised manuscript is essentially ready for publication in Nature Comm.

I have only one comment that should be addressed prior to publication. This same comment was raised in the original review.

1. The N-myristoyl-D-Asn-Leu text in the main text was distracting. The structure could easily be wrong based on the data presented (e.g., N-myristoyl-D-Asn-Ile or something else), and the molecule does not appear to contribute to the biological effects described in the paper for the GABA-containing lipopeptide. If the authors want to keep this data, which is fine, they should downplay this molecule in the main text (which would simplify the additional and somewhat complicated MS text added in the revision) and comment that it is a "predicted metabolite" based on the MS data presented. They should also indicate that Ile is similarly supported by the data. This can be handled at the editorial level and does not require additional reviewer input.

Additional comments:

1. My earlier concerns have been handled in the revised manuscript, especially in regards to the biological comparisons of free GABA versus N-myristoyl-D-Asn-GABA. I didn't believe it based on the original submission data, but with the revised data, I believe this is an exceptional finding. Given that N-acyl-amides are regularly popping up in the human microbiome, this could be a more general strategy for barrier penetration.

2. My earlier worries about metabolite potency and production have also been well handled in the revised manuscript. At this point, I congratulate the authors on a very nice piece of work.

Reviewer #2 (Remarks to the Author):

The authors have performed a large series of additional experiments, which in this reviewer's opinion answers the questions posed to them.

REVIEWERS' COMMENTS:

Reviewer #1 (Remarks to the Author):

The revision is much, much stronger, and in my opinion, the revised manuscript is essentially ready for publication in Nature Comm.

I have only one comment that should be addressed prior to publication. This same comment was raised in the original review.

1. The N-myristoyl-D-Asn-Leu text in the main text was distracting. The structure could easily be wrong based on the data presented (e.g., N-myristoyl-D-Asn-Ile or something else), and the molecule does not appear to contribute to the biological effects described in the paper for the GABA-containing lipopeptide. If the authors want to keep this data, which is fine, they should downplay this molecule in the main text (which would simplify the additional and somewhat complicated MS text added in the revision) and comment that it is a "predicted metabolite" based on the MS data presented. They should also indicate that Ile is similarly supported by the data. This can be handled at the editorial level and does not require additional reviewer input.

The reference to the N-myristoyl-D-Asn-Leu in the text has been downplayed and the sentences have been modified as follow: "(line 170) This common neutral release could correspond to the leucine or the isoleucine (Supplementary Fig. 1 and 3). The proposed interpretation of formation of these product ions are reported in Supplementary Note and Supplementary Fig. 4a."

Additional comments:

1. My earlier concerns have been handled in the revised manuscript, especially in regards to the biological comparisons of free GABA versus N-myristoyl-D-Asn-GABA. I didn't believe it based on the original submission data, but with the revised data, I believe this is an exceptional finding. Given that N-acyl-amides are regularly popping up in the human microbiome, this could be a more general strategy for barrier penetration.

2. My earlier worries about metabolite potency and production have also been well handled in the revised manuscript. At this point, I congratulate the authors on a very nice piece of work.

Reviewer #2 (Remarks to the Author):

The authors have performed a large series of additional experiments, which in this reviewers opinion answers the questions posed to them.